# From Contextual Combinatorial Semi-Bandits to Bandit List Classification: Improved Sample Complexity with Sparse Rewards

**Liad Erez**
Blavatnik School of Computer Science and AI
Tel-Aviv University
liaderez@mail.tau.ac.il

**Tomer Koren**
Blavatnik School of Computer Science and AI
Tel Aviv University and Google Research
tkoren@tauex.tau.ac.il

## Abstract

We study the problem of contextual combinatorial semi-bandits, where input contexts are mapped into subsets of size $m$ of a collection of $K$ possible actions. In each round of the interaction, the learner observes feedback consisting of the realized reward of the predicted actions. Motivated by prototypical applications of contextual bandits, we focus on the $s$-sparse regime where we assume that the sum of rewards is bounded by some value $s \ll K$. For example, in recommendation systems the number of products purchased by any customer is significantly smaller than the total number of available products. Our main result is for the $(\varepsilon, \delta)$-PAC variant of the problem for which we design an algorithm that returns an $\varepsilon$-optimal policy with high probability using a sample complexity of $\widetilde{O}\big((\text{poly}(K/m) + sm/\varepsilon^2) \log(|\Pi|/\delta)\big)$ where $\Pi$ is the underlying (finite) class and $s$ is the sparsity parameter. This bound improves upon known bounds for combinatorial semi-bandits whenever $s \ll K$, and in the regime where $s = O(1)$, the leading terms in our bound match the corresponding full-information rates, implying that bandit feedback essentially comes at no cost. Our PAC learning algorithm is also computationally efficient given access to an ERM oracle for $\Pi$. Our framework generalizes the list multiclass classification problem with bandit feedback, which can be seen as a special case with binary reward vectors. In the special case of single-label classification corresponding to $s = m = 1$, we prove an $O\big((K^7 + 1/\varepsilon^2) \log(|\mathcal{H}|/\delta)\big)$ sample complexity bound for a finite hypothesis class $\mathcal{H}$, which improves upon recent results in this scenario. Additionally, we consider the regret minimization setting where data can be generated adversarially, and establish a regret bound of $\widetilde{O}(|\Pi| + \sqrt{smT \log |\Pi|})$, extending the result of [18] who consider the simpler single label classification setting.

## 1 Introduction

In the contextual combinatorial semi-bandit (CCSB) problem, a learner is tasked with mapping input contexts from a (possibly infinite) context space $\mathcal{X}$ to subsets of a fixed size $m$ of a collection of $K$ available actions. The learner's reward is defined as the sum of the rewards of the predicted actions, and the feedback revealed to the learner consists of the individual reward values of those actions; namely, semi-bandit feedback. The primary objective in this problem is to compete with the best *policy* in an underlying policy class $\Pi$ which is a collection of mappings from $\mathcal{X}$ to subsets of actions of size $m$.

A natural application of this problem, and in fact, one of the classical motivating applications for studying contextual bandits [34] is a recommendation system visited sequentially by users (each arriving with side information playing the role of a context), where upon each visit the system is

39th Conference on Neural Information Processing Systems (NeurIPS 2025).

tasked with presenting a user with a set of $m$ recommended products (or ads) available for purchase, after which the system observes the purchased products as feedback. This is naturally viewed as a bandit scenario in the sense that the system only observes the user's behavior with respect to the recommended products and not others. That is, the only way to obtain feedback on a given product is by actively recommending it to a user.

In such an application, it is natural to assume that each user will only be purchasing up to $s \leq K$ products, where $s$ is considerably smaller than the total number of products $K$, which can be very large. This assumption culminates in a *sparsity* property of the rewards in the underlying combinatorial semi-bandit problem, which raises the question of whether or not sparsity can result in better performance guarantees. More specifically, we are interested in obtaining performance bounds whose leading terms scale primarily with $s$ rather than $K$.

Prior work on combinatorial semi-bandits focuses for the most part on the regret minimization objective, with the rate of $O(\sqrt{mKT})$ obtained by [3] for the vanilla non-contextual variant, and as remarked in the same paper, is optimal for non-sparse losses. The work of [40] provides first-order regret bounds of the form $O(m\sqrt{KL_T^\star})$ where $L_T^\star$ is the cumulative loss of the best arm in hindsight, and while it may be a significant improvement in cases where the cumulative loss of the best arm is sublinear in $T$, the bound still contains a polynomial dependence on $K$. To our knowledge, no prior works on either the vanilla or the contextual variants prove regret bounds which scale with the sparsity $s$ of the losses instead of directly with $K$.

An important special case of CCSB is the (agnostic) *bandit multiclass list classification* problem, in which contexts, actions and policies correspond to examples, labels and hypotheses, respectively. In this CCSB variant, the rewards exhibit a special structure; namely, the reward of a predicted label is the zero-one reward. Existing work on multiclass list classification focuses on the full-information single-label setting, where a given example has a single correct label which is revealed to the learner after each prediction. The primary focus on this literature is on characterizing properties of the underlying hypothesis class $\mathcal{H}$ under which various notions of learnability can be achieved; e.g., PAC learnability [10], uniform convergence [24] and online learnability [39]. In bandit multiclass list classification, upon predicting a list of labels of size $m \leq K$, the learner observes partial, a.k.a. "bandit" feedback consisting only of the predicted labels which belong to the collection of $s$ correct labels for the given example.[1]

In the context of traditional multiclass classification (that is, when $m = s = 1$), extensive research has studied the bandit variant of the problem, starting with the work of [30] who consider linear classification. Follow-up works study various questions of learnability of general hypothesis classes [15, 13, 44, 19] and other recent works [17, 18] study optimal rates for both the regret minimization and PAC objectives with a focus on finite hypothesis classes. It is thus natural to study the extension of the problem to list multiclass classification with bandit feedback which, to our knowledge, our work is the first to tackle.

For contextual bandits ($m = 1$) with finite policy classes $\Pi$, the recent work of [18] characterized the optimal *regret*; that is, the cumulative reward of the learner compared to that of the best policy in $\Pi$, and showed that it is of the form $\Theta(\min\{|\Pi| + \sqrt{sT}, \sqrt{KT \log |\Pi|}\})$, implying that for relatively small policy classes the classical $\sqrt{KT}$ rate of EXP4 [5] can be improved if $s \ll K$. In a subsequent work, [17] study the PAC objective in single-label bandit multiclass classification ($s = m = 1$) where the goal is to learn a near-optimal hypothesis with respect to some unknown data distribution, and establish a sample complexity bound of $O((K^9 + 1/\varepsilon^2) \log(|\mathcal{H}|/\delta))$ for $(\varepsilon, \delta)$-PAC learning in the single-label setting using a computationally efficient algorithm given an ERM oracle to $\mathcal{H}$. One of the key takeaways from both of these works is that in the single-label setting, bandit feedback essentially comes at no additional cost compared to full feedback in the asymptotic regimes where $T \to \infty$ and $\varepsilon \to 0$.

We are thus motivated to investigate the following more general question: *what is the role of reward sparsity in contextual combinatorial semi-bandits?* In this work, we provide an answer to this question by designing PAC learning and regret minimization algorithms for CCSB with finite policy classes, and prove that they attain sample complexity and regret bounds whose dominant terms scale with the sparsity parameter $s$ rather than $K$.

---

[1]In the wider context of CCSB, this feedback actually corresponds to *semi-bandit feedback*, as defined earlier.

To illustrate the challenge in answering this question, let us consider the naive approach for PAC learning in CCSB by which subsets of actions of size $m$ are sampled uniformly at random, and bandit feedback is used to estimate the rewards of policies in $\Pi$ via importance sampling. Since the variance of such reward estimators scales polynomially with $K$, this approach ultimately results in a sample complexity of $\approx Km/\varepsilon^2$. This bound is far from optimal since, as we will show, the leading term in the optimal sample complexity bound in fact scales with $s$ instead of $K$.

## 1.1 Summary of Contributions

We summarize our results for contextual combinatorial semi-bandits over a finite policy class $\Pi$:

- Our main result involves the PAC setting (see Section 3), where we design an algorithm that with probability at least $1 - \delta$ outputs a policy $\pi_{\text{out}} \in \Pi$ that is $\varepsilon$-optimal with respect to the population reward, using a sample complexity of at most

$$O\left(\left(\text{poly}\left(\frac{K}{m}\right) + \frac{sm}{\varepsilon^2}\right)\log\frac{|\Pi|}{\delta}\right).$$

  Moreover, our algorithm is computationally efficient given an ERM oracle for $\Pi$. We also present the high-level construction of a sample complexity lower bound of $\Omega(sm/\varepsilon^2)$ which holds even for a single context, showing that the dominant term in our bound is in fact optimal. As an immediate corollary, the above results hold in the special case of *bandit multiclass list classification* with $s$-sparse rewards over a finite hypothesis class.

- In the special case of single label bandit multiclass classification corresponding to $s = m = 1$, we show that a slight variation of our approach results in a sample complexity of $O\left((K^7 + 1/\varepsilon^2)\log(|\mathcal{H}|/\delta)\right)$, which has an improved dependence on $K$ compared to the existing bounds in [17].

- We also consider the online regret minimization setting, where we design an algorithm achieving an expected regret bound of $\tilde{O}(|\Pi| + \sqrt{smT\log|\Pi|})$, which holds even when the input and reward sequence is adversarial. See Appendix B for more details.

Our starting point for addressing the PAC setting in CCSB is the recent work of [17] who study the single-label ($s = m = 1$) classification setting. Following the general scheme they proposed, our algorithm operates in two stages by first computing a low-variance exploration distribution over policies in $\Pi$ and then using this distribution to uniformly estimate the policies' expected rewards. Generalizing the single-label classification setting to CCSB, however, comes with some nontrivial technical challenges. First, [17] use the single-label assumption in order to collect a labeled dataset by uniformly sampling labels. In the general setting, however, this approach cannot work effectively, since the full reward function cannot be inferred unless the predicted set contains all $s$ correct labels, which is a very unlikely event. We circumvent the issue by introducing additional importance sampling estimation with an appropriate modification of the convex potential in use. Moreover, as a means to compute a low-variance exploration distribution, rather than using a stochastic optimization scheme as suggested in [17], we optimize an empirical objective and use Rademacher complexity arguments in order to establish its uniform convergence to the expected objective given sufficient samples. This approach allows us in particular to improve the overall dependence on $K$ in the single-label classification setting, leveraging refined $L_\infty$ vector-contraction results for the Rademacher complexity [21].

## 1.2 Additional Related Work

**Combinatorial semi-bandits.**  The non-contextual variant of the combinatorial semi-bandit problem was introduced by [23] in the framework of online shortest paths. This problem has been extensively studied in the bandit literature, mostly in the context of regret minimization [3, 48, 33, 40, 47, 27, etc.]. A regret bound of $O(\sqrt{mKT})$ was shown by [3] for adversarial losses, and was proven by [36] to be optimal in the multiple-play setting in which the available combinatorial actions are all subsets of size $m$ of the action set (which is the setting considered in this paper). The contextual combinatorial semi-bandit (CCSB) problem is considerably less explored, with some existing works [43, 48, 46, 49] focusing on the case where rewards are noisy linear functions of the inputs, and others [31, 32] which consider a setting similar to ours with finite unstructured policy classes, and

establish regret bounds of the form $O(\sqrt{mKT \log |\Pi|})$. To our knowledge, all previous results on combinatorial semi-bandits exhibit a polynomial dependence on $K$ in the leading term, thus leaving open the question of adaptivity to reward sparsity.

**Combinatorial (full-)bandits.** Another well-studied variant of CCSB is known as *combinatorial bandits* (also referred to as *bandit combinatorial optimization*, [38, 6]), where the feedback is limited only to the realized reward, that is, the sum of rewards of predicted actions. For the non-sparse version of the problem, regret bounds of $O(m^{3/2}\sqrt{KT})$ have been established in several previous works [12, 1, 8, 9, 25] and have subsequently been shown to be optimal [11, 28]. In the context of multiclass classification, while perhaps not as natural as semi-bandit feedback, full-bandit feedback is well-motivated in some applications, for example, in cases where a recommendation system is interested in protecting the users' privacy by only observing the amount of products purchased rather than the products themselves. While our results apply in the semi-bandit model, examining the full-bandit variant raises some very interesting questions and generalizing our results to this setting seems highly non-trivial; see Section 4 for further discussion.

**List multiclass Classification.** The theoretical framework of multiclass list classification was originally introduced by [7]. [10] provide a characterization of PAC learnability for multiclass list classification by generalizing the DS-dimension [14], [39] consider the regret minimization setting and characterize learnability by a generalization of the Littlestone dimension, and [24] study the notions of uniform convergence and sample compression in the context of multiclass list classification. A regression-variant of list learning has also been studied in [42] who provide a characterization of learnability for this problem. Notably, all of these works on multiclass list classification focus on the single-label setting.

**Bandit multiclass classification.** The setting of bandit multiclass classification was originally introduced by [30], with [15] showing that learnability of deterministic learners in the realizable setting is characterized by the bandit Littlestone dimension. [13] generalize those results by showing that the bandit Littlestone dimension characterizes online learnability whenever the label set is finite, and [44] generalize this result to infinite label sets. Several previous works [4, 15, 37] study the price of bandit feedback in the realizable setting, with the recent work of [19] showing that this price is bounded by a factor of $O(K)$ over the mistake bound in the full-information setting for randomized learners.

## 2  Problem Setup

We study a learning scenario where a learner has to map contexts from a domain $\mathcal{X}$ to subsets of size $m$ of a collection of $K \geq m$ possible actions, denoted by $\mathcal{Y} \triangleq [K]$. We denote by $\mathcal{A} \triangleq \{a \in \{0,1\}^K \mid \|a\|_1 = m\}$,[2] the set of available combinatorial actions. In the semi-bandit setting, the learner iteratively interacts with an environment according to the following, for $t = 1, 2, \ldots$:

  (i) The environment generates a context-reward vector pair $(x_t, r_t)$ where $x_t \in \mathcal{X}$ and $r_t \in [0, 1]^K$, the learner receives $x_t$;
 (ii) The learner predicts $a_t \in \mathcal{A}$;
(iii) The learner gains reward $r_t \cdot a_t$ (namely the sum of rewards of predicted actions) and observes *semi-bandit feedback* consisting of the rewards of the predicted actions $\{r_t(y) \mid y \in a_t\}$.

We assume that there is a known bound $s \leq K$ on the $L_1$-norm of all reward vectors, that is, $\|r_t\|_1 \leq s$ for all $t$.[3] We remark that since $r_t(\cdot) \in [0, 1]$, this sparsity condition also implies that $\|r_t\|_2^2 \leq s$. In the settings we consider, the learner's performance is measured with respect to an underlying policy class $\Pi \subseteq \{\mathcal{X} \to \mathcal{A}\}$; focusing in this work on the case where $\Pi$ is finite.

**PAC setting.** In the $(\varepsilon, \delta)$-PAC setting, the context-reward pairs $(x_t, r_t)$ are generated in an i.i.d manner by some unknown distribution $\mathcal{D}$. The learner's goal is to compute, using as few samples as possible, a policy $\pi_{\text{out}} \in \Pi$ such that with probability at least $1 - \delta$ (over all randomness during the

---

[2]In some formulations, the available actions come from a fixed, arbitrary subset of $\mathcal{A}$; here we focus on the setting where all subsets of $\mathcal{Y}$ of size $m$ are valid.

[3]This notion of sparsity is weaker than strict sparsity, where the rewards have at most $s$ nonzero coordinates.

interaction with the environment),

$$r_{\mathcal{D}}(\pi^{\star}) - r_{\mathcal{D}}(\pi_{\text{out}}) \le \varepsilon,$$

where $r_{\mathcal{D}}(\pi) = \mathbb{E}_{(x,r)\sim\mathcal{D}}[r^{\top}\pi(x)]$ and $\pi^{\star} = \operatorname{argmax}_{\pi\in\Pi} r_{\mathcal{D}}(\pi)$ is the population reward of $\pi$, and $\pi^{\star} \triangleq \operatorname{argmax}_{\pi\in\Pi} r_{\mathcal{D}}(\pi)$ is the best policy in the class $\Pi$ w.r.t. $\mathcal{D}$. The learner's performance in this setting is measured in terms of *sample complexity*, that is, the number of samples $(x_t, r_t)$ generated by the environment during the interaction as a function of $(\varepsilon, \delta)$ after which the learner outputs a policy $\pi_{\text{out}}$ satisfying the guarantee above.

**Regret minimization setting.** In the online regret setting, the pairs $(x_t, r_t)$ are generated by an oblivious adversary.[4] The interaction lasts for $T$ rounds, where in each round $t$, after predicting $a_t \in \mathcal{A}$, the learner gains a reward $r_t(a_t) \triangleq r_t \cdot a_t \in [0, s]$, and the objective is to ultimately minimize *regret* with respect to $\Pi$, defined as

$$\mathcal{R}_T \triangleq \sup_{\pi^{\star}\in\Pi} \sum_{t=1}^{T} r_t^{\top}\pi^{\star}(x_t) - \sum_{t=1}^{T} r_t^{\top}a_t.$$

**ERM oracle.** To discuss the computational efficiency of our PAC learning algorithm, we assume access to an empirical risk minimization (ERM) oracle for $\Pi$. This oracle, denoted by $\mathsf{ERM}_{\Pi}$, gets as input a collection of pairs $S = \{(x_1, \hat{r}_1), \dots, (x_n, \hat{r}_n)\} \subseteq \mathcal{X} \times \mathbb{R}_+^K$ and returns

$$\mathsf{ERM}_{\Pi}(S) \in \operatorname{argmax}_{\pi\in\Pi} \sum_{i=1}^{n} \sum_{y\in\pi(x_i)} \hat{r}_i(y).$$

This is a natural generalization of the optimization oracle used in previous works on contextual bandits [16, 2]. When we refer to the computational complexity of our algorithm, we assume that each call to $\mathsf{ERM}_{\Pi}$ takes constant time.

**Bandit Multiclass List Classification.** The semi-bandit multiclass list classification problem is a special case of CCSB, with the following specialized notation: The set of all lists (subsets) of $\mathcal{Y}$ of size $m$ is denoted by $\mathcal{L}$, and instead of a policy class we refer to a *hypothesis class* $\mathcal{H} \subseteq \{\mathcal{X} \to \mathcal{L}\}$. In every round $t$ of the interaction, the environment generates a pair $(x_t, Y_t)$ where $x_t \in \mathcal{X}$ and $Y_t \subseteq \mathcal{Y}$ where $Y_t$ corresponds to the set of all true labels at round $t$. We assume that $|Y_t| \le s$ for all $t$ for some known bound $s$ which corresponds to reward sparsity in this setting. Upon predicting a list $L_t \in \mathcal{L}$, the learner observes semi-bandit feedback consisting of $L_t \cap Y_t$, which corresponds to the zero-one reward values for all labels in $a_t$.

## 3 Main Result: Agnostic PAC Setting

In this section we design and analyze a PAC learning algorithm for CCSB with $s$-sparse rewards. Our algorithm is displayed in Algorithm 1, and our main result is detailed in the following theorem:

**Theorem 1.** *If we set* $\gamma = \frac{1}{2}$, $N_1 = \widetilde{\Theta}(\frac{K^9}{m^8}\log(|\Pi|/\delta)))$, $N_2 = \Theta\big((K/m\varepsilon + sm/\varepsilon^2)\log(|\Pi|/\delta)\big)$, $T = \Theta((K/m)^5)$, *then with probability at least* $1 - \delta$ *Algorithm 1 outputs* $\pi_{\text{out}} \in \Pi$ *with* $r_{\mathcal{D}}(\pi^{\star}) - r_{\mathcal{D}}(\pi_{\text{out}}) \le \varepsilon$ *using a sample complexity of*

$$N_1 + N_2 = O\left(\left(\frac{K^9}{m^8} + \frac{sm}{\varepsilon^2}\right)\log\frac{|\Pi|}{\delta}\right).$$

*Furthermore, Algorithm 1 makes a total of* $T + 1 = O((K/m)^5)$ *calls to* $\mathsf{ERM}_{\Pi}$.

In Appendix A.3 we show that in the special case of single-label classification (corresponding to $s = m = 1$ and zero-one rewards), a specialized version of Algorithm 1 results in a sample complexity bound of

$$O\left(\left(K^7 + \frac{1}{\varepsilon^2}\right)\log\frac{|\mathcal{H}|}{\delta}\right),$$

---

[4]We consider an oblivious adversary throughout, though most of our results extend to an adaptive adversary.

---

**Algorithm 1** PAC-COMBAND

---

Parameters: $N_1, N_2, \gamma \in (0, \frac{1}{2}], T \in \mathbb{N}$.
**Phase 1:**
**for** $i = 1, \ldots, N_1$ **do**
    // Environment generates $(x_i, r_i) \sim \mathcal{D}$, algorithm receives $x_i$.
    Predict $a_i \in \mathcal{A}$ uniformly at random and receive feedback $\{r_i(y) \mid y \in a_i\}$.
**end for**
Initialize $p_1 \in \Delta_\Pi$ to be a delta distribution.
**for** $t = 1, \ldots, T$ **do**
    Let $q_t \in \Delta_\Pi$ be the delta distribution on $\pi_t = \mathsf{ERM}_\Pi\left(\{(x_i, \hat{r}_i)\}_{i=1}^{N_1}\right)$, where

$$\hat{r}_i(y) = \frac{1}{N_1}(1 - \gamma)\frac{K}{m}\frac{\mathbb{1}\{y \in a_i\}r_i(y)}{Q_{x_i,y}^{\gamma}(p_t)}, \quad \forall i \in [N_1], y \in \mathcal{Y}.$$

    Update $p_{t+1} = (1 - \eta_t)p_t + \eta_t q_t$ where $\eta_t = 2/(2 + t)$.
**end for**
Let $\hat{p} = p_{T+1}$.
**Phase 2:**
**for** $i = 1, \ldots, N_2$ **do**
    // Environment generates $(x_i, r_i) \sim \mathcal{D}$, algorithm receives $x_i$.
    With prob. $\gamma$ pick $a_i \in \mathcal{A}$ uniformly at random; otherwise sample $\pi_i \sim \hat{p}$ and set $a_i = \pi_i(x_i)$.
    Predict $a_i$ and receive feedback $\{r_i(y) \mid y \in a_i\}$.
**end for**
Return:

$$\pi_{\text{out}} = \mathsf{ERM}_\Pi\left(\{(x_i, \hat{r}_i)\}_{i=1}^{N_2}\right), \quad \text{where} \quad \hat{r}_i(y) = \frac{\mathbb{1}\{y \in a_i\}r_i(y)}{Q_{x_i,y}^{\gamma}(\hat{p})} \quad \forall y \in \mathcal{Y}.$$

---

which has an improved dependence on $K$ compared to the bound obtained by [17] which scales as $K^9$.

Given a context-action pair $(x, y) \in \mathcal{X} \times \mathcal{Y}$ and $p \in \Delta_\Pi \triangleq \left\{p \in \mathbb{R}_+^{|\Pi|} \mid \sum_{i=1}^{|\Pi|} p_i = 1\right\}$ we denote

$$Q_{x,y}(p) \triangleq \sum_{\pi \in \Pi} p(\pi)\mathbb{1}\{y \in \pi(x)\},$$

i.e., the probability that $y$ belongs to $\pi(x)$ when sampling $\pi \sim p$, and given $\gamma \in (0, 1)$ we define

$$Q_{x,y}^{\gamma}(p) \triangleq (1 - \gamma)Q_{x,y}(p) + \gamma m/K,$$

that is, the distribution induced by mixing $p$ with a uniform distribution over $\mathcal{A}$.

At a high level, our algorithm initially uses the Frank-Wolfe algorithm to approximately solve the following convex optimization problem:

$$\text{minimize} \quad \widehat{F}(p) \triangleq \frac{1}{N_1}\sum_{i=1}^{N_1} f(p; z_i), \quad p \in \Delta_\Pi, \tag{1}$$

$$\text{where} \quad f(p; z = (x, r, a)) \triangleq -\frac{K}{m}\sum_{y \in a} r(y)\log(Q_{x,y}^{\gamma}(p)). \tag{2}$$

Here $\mathcal{D}'$ is the product distribution over $\mathcal{Z} \triangleq \mathcal{X} \times [0, 1]^K \times \mathcal{A}$ defined as $\mathcal{D}' \triangleq \mathcal{D} \times \mathrm{Unif}(\mathcal{A})$, that is, a pair $(x, r) \in \mathcal{X} \times [0, 1]^K$ is sampled from $\mathcal{D}$ and $a \in \mathcal{A}$ is sampled *independently* uniformly at random. The random objective $f(\cdot; z)$ is defined according to Eq. (2), and we note that even though the full reward function $r$ is not observed, $f(\cdot; z)$ depends only on the reward of the actions in $a \sim \mathcal{A}$ and can thus be fully accessed. The $K/m$ factor in Eq. (2) is necessary due to the random sampling of $a \sim \mathcal{A}$ used for importance-weighted estimation of the reward function, and it is straightforward to

see that the expected objective has the following form:

$$F(p) \triangleq \mathbb{E}_{z \sim \mathcal{D}'}[f(p;z)] = \mathbb{E}_{(x,r) \sim \mathcal{D}}\left[ -\sum_{y \in \mathcal{Y}} r(y) \log\big(Q_{x,y}^{\gamma}(p)\big) \right]. \tag{3}$$

This form of $F$ is of crucial importance as its gradient at a point $p$ is proportional to the variance of an appropriate unbiased reward estimator for the policies in $\Pi$:

$$\text{Var}\big[R_p(\pi_j)\big] \lesssim m \cdot \left| \frac{\partial F}{\partial p_j}(p) \right| \quad \forall \pi_j \in \Pi.$$

As we will show, $\text{poly}(K/m)$ samples suffice in order for the empirical objective $\widehat{F}$ to approximate the expected objective $F(\cdot)$ with sufficient accuracy *uniformly* over $\Delta_\Pi$, resulting in the fact that an approximate minimizer $\hat{p} \in \Delta_\Pi$ of $\widehat{F}(\cdot)$ satisfies $\|\nabla F(\hat{p})\|_\infty \lesssim s$. This fact generalizes the key insight from the previous works of [16, 2, 17] in the sense that the reward estimators induced by $\hat{p}$ have variance bounded by $C \cdot sm$, where $C$ is an absolute constant. Crucially, this variance bound doesn't depend directly on the number of actions $K$, implying that we can use $\hat{p}$ to estimate the expected rewards of the policies in $\Pi$ with only $\approx sm/\varepsilon^2$ samples by Bernstein's variance-sensitive concentration inequality.

## 3.1 Analysis

Here we detail the main steps in the analysis of Algorithm 1, and in particular, outline the main challenges compared to the single-label setting where $s = m = 1$.

**Initial exploration.** While in the single label classification setting it is possible to collect a dataset containing $\text{poly}(K)$ i.i.d samples from $\mathcal{D}$ by simply predicting labels uniformly, this approach does not work in the multilabel classification setting (i.e. $s > 1$) and neither in CCSB with $s$-sparse rewards. The reason is that in these settings a uniform prediction of a list of actions will simply not yield a full observation of the reward vector, even after $\text{poly}(K)$ such predictions. Thus, instead of collecting a dataset, we use semi-bandit feedback directly in order to estimate the convex objective of interest using importance sampling, as detailed in Algorithm 1. These random objectives are unbiased with respect to $F(\cdot)$ defined in Eq. (3), while importance weighting adds a scaling factor of $K/m$. An additional noteworthy difference in our approach is the fact that rather than using a stochastic optimization procedure to minimize the stochastic objective in Eq. (3) directly, we optimize the empirical version of this objective and use a uniform convergence argument to show that an approximate empirical minimizer also achieves nearly optimal expected function value. Our uniform convergence analysis of the underlying function class leverages $L_\infty$ vector-contraction properties of Rademacher complexities introduced in [20]; see Appendix A.1 for more details.

**Optimizing for low-variance exploration.** In order to approximately solve the convex optimization problem defined in Eq. (1), we employ the Frank-Wolfe (FW) algorithm [22], and make use of a convergence result by [29] which allows us to take advantage of $L_1/L_\infty$-smoothness properties of the objective. In more detail, we run Frank-Wolfe on the objective defined in Eq. (1) over samples generated by the distribution $\mathcal{D}'$, where crucially the gradients of the random objectives defined in Eq. (2) can be calculated exactly via

$$\frac{\partial f(p;x,r,a)}{\partial p_j} = -(1-\gamma)\frac{K}{m}\sum_{y \in a}\frac{\mathbb{1}\{y \in \pi_j(x)\}r(y)}{Q_{x,y}^{\gamma}(p)}, \quad \forall j \in [|\Pi|]. \tag{4}$$

The FW algorithm uses these gradients with a linear optimization oracle $\mathsf{LOO}_\Pi$, defined by

$$\mathsf{LOO}_\Pi(v) \triangleq \underset{p \in \Delta_\Pi}{\arg\min}\, v \cdot p, \quad \forall v \in \mathbb{R}^{|\Pi|}.$$

Importantly, each call to $\mathsf{LOO}_\Pi$ can be implemented by a call to $\mathsf{ERM}_\Pi$ (see Appendix A.2 for the proof). The analysis of [29] applied to the objective in Eq. (1) shows that with an appropriate choice of parameters, the Frank-Wolfe algorithm outputs a $\mu$-approximate minimizer of Eq. (1) after $T = O(sK^2/(\gamma m^2 \mu))$ iterations. This follows from $L_1$-smoothness properties of the functions defined in Eq. (2) with smoothness parameter $\beta = sK^3/(\gamma^2 m^3)$. For more details, see Appendix A.2.

**Low-variance reward estimation.**   In the second phase of Algorithm 1, we use the low-variance exploration distribution $\hat{p} \in \Delta_\Pi$ computed in the first phase in order to estimate the expected reward of all policies in $\Pi$ using importance-weighted estimators, defined as

$$R_i(\pi) \triangleq \sum_{y \in \pi(x)} \frac{r_i(y)\mathbb{1}\{y \in a_i\}}{Q^\gamma_{x_i,y}(\hat{p})}, \quad i \in [N_2].$$

The guarantee on $\hat{p}$ provides us with the ability to bound the variance of these estimators by $O(sm)$ (with no explicit dependence on $K$) which is why, using Bernstein's inequality, $O(K/(m\varepsilon) + sm/\varepsilon^2)$ samples in the second phase are sufficient for the average estimated rewards to constitute $\varepsilon$-approximations of the true rewards uniformly over all policies in $\Pi$, thus implying the PAC guarantee for the policy which maximizes the average estimated reward.

*Proof of Theorem 1 (sketch).*  We now give an overview of the proof of Theorem 1. We rely on the following key lemma which shows that a sufficiently approximate optimum $\hat{p}$ of the convex objective defined in Eq. (3) is a point at which the gradient is bounded in $L_\infty$ norm by $s$. The proof of this lemma can be found in Appendix A.  □

**Lemma 1.** *Suppose $\gamma \leq \frac{1}{2}$ and let $\hat{p} \in \Delta_\Pi$ be an approximate minimizer of the objective $F(\cdot)$ defined in Eq. (3) up to an additive error of $\mu$. Then,*

$$\|\nabla F(\hat{p})\|_\infty \leq s + \sqrt{\frac{2s\mu K^2}{\gamma^2 m^2}}.$$

*In particular, setting $\mu = s\gamma^2 m^2/2K^2$ gives $\|\nabla F(\hat{p})\|_\infty \leq 2s$.*

In order to apply Lemma 1, we make use of a uniform convergence argument which implies that given sufficiently many samples, an approximate minimizer of the empirical objective defined in Eq. (1) is also an approximate minimizer of the stochastic objective defined in Eq. (3). This lemma is also proven in Appendix A.

**Lemma 2.** *Assume $N_1 = \widetilde{\Theta}\big(\frac{s^2 K^5}{m^4 \mu^2} \log \frac{|\Pi|}{\delta}\big)$ and that phase 1 of Algorithm 1 results in $\hat{p} \in \Delta_\Pi$ which minimizes $\widehat{F}(\cdot)$ defined in Eq. (1) up to an additive error of $\mu/3$. Then with probability at least $1 - \delta$, $\hat{p}$ minimizes $F(\cdot)$ defined in Eq. (3) up to an additive error of $\mu$.*

In Appendix A.2 we prove that $T = O(sK^3/(\gamma^2 m^3 \mu))$ iterations of FW result in $\hat{p} \in \Delta_\Pi$ that minimizes $\widehat{F}(\cdot)$ up to accuracy $\mu/3$. Setting $\mu = s\gamma^2 m^2/2K^2$, Lemma 1 and Lemma 2 imply that phase 1 of Algorithm 1 yields a distribution $\hat{p} \in \Delta_\Pi$ satisfying

$$\mathbb{E}_{(x,r) \sim \mathcal{D}}\left[ \sum_{y \in \pi(x)} \frac{r(y)}{Q^\gamma_{x,y}(\hat{p})} \right] \leq 4s \quad \forall \pi \in \Pi. \tag{5}$$

In phase 2, Algorithm 1 uses samples from $\hat{p}$ to estimate the expected rewards of policies in $\Pi$ with the following estimators:

$$R_i(\pi) \triangleq \sum_{y \in \pi(x)} \frac{r_i(y)\mathbb{1}\{y \in a_i\}}{Q^\gamma_{x_i,y}(\hat{p})}, \quad i \in [N_2].$$

It is straightforward to see that this is an unbiased estimator for $r_\mathcal{D}(\pi)$, and moreover, using the Cauchy-Schwarz inequality and Eq. (5), its variance can be upper bounded by

$$\mathrm{Var}[R_i(\pi)] \leq m\mathbb{E}\left[ \sum_{y \in \pi(x_i)} \frac{r_i(y)}{Q^\gamma_{x_i,y}(\hat{p})} \right] \leq 4sm.$$

Using Bernstein's inequality and the fact that the random variables $R_i(\pi)$ are bounded by $K/(\gamma m)$ we deduce that the following sample complexity suffices for $(\varepsilon, \delta)$-PAC:

$$N_1 + N_2 = \Theta\left( \left( \frac{K^9}{m^8} + \frac{K}{m\varepsilon} + \frac{sm}{\varepsilon^2} \right) \log \frac{|\Pi|}{\delta} \right),$$

and second term can be dropped since the sum of the other two terms dominates the bound.

## 3.2 Lower Bound

We conclude this section with presenting a lower bound, according to which the dependence on $m$ and $s$ in Theorem 1 is tight even in the non-contextual setting.

**Theorem 2.** *For any combinatorial semi-bandit algorithm* Alg *over $K$ actions and combinatorial actions of size $m \leq K/2$, there exists an $s$-sparse instance for which in order for* Alg *to output $\hat{a} \in \mathcal{A}$ which is $\varepsilon$-optimal for $\varepsilon \ll 1/K$ with constant probability,* Alg *requires a sample complexity of at least $\widetilde{\Omega}(sm/\varepsilon^2)$.*

We provide a sketch of the proof of Theorem 2 below, see the formal statement and proof in Appendix A.4.

*Proof of Theorem 2 (sketch).* Consider an instance specified by $m$ Bernoulli arms with expected reward $\frac{s}{K} + \frac{\varepsilon}{m}$ each while the other $K - m$ arms have expected reward $\frac{s}{K} - \frac{\varepsilon}{K-m}$. Denote the set of $m$ arms with highest expected reward by $\mathcal{S}$. Now, note that in order to find an $\frac{\varepsilon}{2}$-optimal subset, Alg must identify at least half of the arms in $\mathcal{S}$, and thus is required to estimate all of the arms' rewards up to an accuracy of $\frac{\varepsilon}{2m}$. Since each arm's reward distribution has variance of $\approx \frac{s}{K}$, the number of samples required to make such an estimation of the reward for each arm $y$ is $\frac{\text{Var}(r_y)}{(\varepsilon/m)^2} \approx \frac{s}{K} \cdot \frac{m^2}{\varepsilon^2}$. Since such an estimation needs to be performed for all $K$ arms and each time step gives Alg samples for $m$ arms via semi-bandit feedback, the total sample complexity is of order at least

$$\frac{K}{m} \cdot \frac{s}{K} \cdot \frac{m^2}{\varepsilon^2} = \frac{sm}{\varepsilon^2}.$$

□

## 4 Conclusion and Open Problems

In this work, we provide sample complexity bounds for contextual combinatorial semi-bandits whose dominant terms scale with an underlying sparsity parameters $s$ rather than with the number of actions $K$. We also give regret bounds for the online (adversarial) version of the problem. Our results apply to the special case of bandit multiclass list classification, and constitute a generalization of previous work on the single-label classification setting [17, 18].

Our works leaves several compelling open questions, such as the extension of our results to the full-bandit setting, improving the dependence on $K$ in the sample complexity bound, and extending our results to infinite policy classes. For further discussion of future directions, see Appendix C.

## Acknowledgments

This project has received funding from the European Research Council (ERC) under the European Union's Horizon 2020 research and innovation program (grant agreement No. 101078075). Views and opinions expressed are however those of the author(s) only and do not necessarily reflect those of the European Union or the European Research Council. Neither the European Union nor the granting authority can be held responsible for them. This work received additional support from the Israel Science Foundation (ISF, grant numbers 2549/19 and 3174/23), a grant from the Tel Aviv University Center for AI and Data Science (TAD) and from the Len Blavatnik and the Blavatnik Family foundation.

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

# A Proofs for Section 3

To prove Lemma 1, we will need the following auxiliary result:

**Lemma 3.** *Suppose $\gamma \leq \frac{1}{2}$ and let $p^\star = \operatorname{argmin}_{P \in \Delta_\Pi} F(P)$. Then it holds that $\|\nabla F(p^\star)\|_\infty \leq s$.*

*Proof.* First note that the gradient of $F(\cdot)$ is given by

$$(\nabla F(p))_\pi = \mathbb{E}\left[-\sum_{y \in \mathcal{Y}} \frac{(1-\gamma)r(y)\mathbb{1}\{y \in \pi(x)\}}{Q_{x,y}^\gamma(p)}\right].$$

Thus, using first-order convex optimality conditions, the following holds that for any $p \in \Delta_\Pi$,

$$\nabla F(p^\star) \cdot (p - p^\star) \geq 0,$$

which with the explicit form of the gradient becomes

$$\mathbb{E}\left[\sum_{y \in \mathcal{Y}} \frac{(1-\gamma)r(y)\sum_{\pi \in \Pi}(p(\pi) - p^\star(\pi))\mathbb{1}\{y \in \pi(x)\}}{Q_{x,y}^\gamma(p^\star)}\right] \leq 0,$$

or equivalently, after dividing by $1 - \gamma$,

$$\mathbb{E}\left[\sum_{y \in \mathcal{Y}} \frac{r(y)(Q_{x,y}^\gamma(p) - Q_{x,y}^\gamma(p^\star))}{Q_{x,y}^\gamma(p^\star)}\right] \leq 0,$$

which rearranges to

$$\mathbb{E}\left[\sum_{y \in \mathcal{Y}} \frac{r(y)Q_{x,y}^\gamma(p)}{Q_{x,y}^\gamma(p^\star)}\right] \leq \mathbb{E}\left[\sum_{y \in \mathcal{Y}} r(y)\right] = \mathbb{E}[\|r\|_1] \leq s.$$

Letting $p \in \Delta_\Pi$ be the distribution concentrated at some $\pi \in \Pi$, we get

$$\mathbb{E}\left[(1-\gamma)\sum_{y \in \mathcal{Y}} \frac{r(y)\mathbb{1}\{y \in \pi(x)\}}{Q_{x,y}^\gamma(p^\star)}\right] \leq s,$$

and the claim follows since the left-hand side equals $(-\nabla F(p^\star))_\pi$ and is nonnegative. $\qquad\square$

*Proof of Lemma 1.* Using Lemma 3 and the triangle inequality, it suffices to show that

$$\|\nabla F(\hat{p}) - \nabla F(p^\star)\|_\infty \leq \sqrt{\frac{2s\mu K^2}{\gamma^2 m^2}},$$

and by the assumption on $\hat{p}$ this clearly holds if

$$\|\nabla F(\hat{p}) - \nabla F(p^\star)\|_\infty^2 \leq \frac{2sK^2}{\gamma^2 m^2}(F(\hat{p}) - F(p^\star)).$$

To prove this inequality, we start with the left-hand side and use the explicit form of $\nabla F(\cdot)$:

$$\|\nabla F(\hat{p}) - \nabla F(p^\star)\|_\infty^2 = \max_{\pi \in \Pi}(1-\gamma)^2 \left( \mathbb{E}\left[ \sum_{y \in \mathcal{Y}} \frac{r(y)\mathbb{1}\{y \in \pi(x)\}}{Q_{x,y}^\gamma(\hat{p})} \right] - \mathbb{E}\left[ \sum_{y \in \mathcal{Y}} \frac{r(y)\mathbb{1}\{y \in \pi(x)\}}{Q_{x,y}^\gamma(p^\star)} \right] \right)^2$$

$$\leq \mathbb{E}\left( \sum_{y \in \mathcal{Y}} \frac{r(y)}{Q_{x,y}^\gamma(\hat{p})} - \sum_{y \in \mathcal{Y}} \frac{r(y)}{Q_{x,y}^\gamma(p^\star)} \right)^2$$

$$= \mathbb{E}\left[ \|r\|_1^2 \left( \sum_{y \in \mathcal{Y}} \frac{r(y)}{\|r\|_1} \left( \frac{1}{Q_{x,y}^\gamma(\hat{p})} - \frac{1}{Q_{x,y}^\gamma(p^\star)} \right) \right)^2 \right]$$

$$\leq \mathbb{E}\left[ \|r\|_1 \sum_{y \in \mathcal{Y}} r(y) \left( \frac{1}{Q_{x,y}^\gamma(\hat{p})} - \frac{1}{Q_{x,y}^\gamma(p^\star)} \right)^2 \right]$$

$$\leq s \cdot \mathbb{E}\left[ \sum_{y \in \mathcal{Y}} r(y) \left( \frac{1}{Q_{x,y}^\gamma(\hat{p})} - \frac{1}{Q_{x,y}^\gamma(p^\star)} \right)^2 \right], \tag{6}$$

where we used Jensen's inequality twice and the fact that $\|r\|_1 \leq s$. Now, for any $(x,y) \in \mathcal{X} \times \mathcal{Y}$ it holds that

$$\left( \frac{1}{Q_{x,y}^\gamma(\hat{p})} - \frac{1}{Q_{x,y}^\gamma(p^\star)} \right)^2 = \frac{(Q_{x,y}^\gamma(\hat{p}) - Q_{x,y}^\gamma(p^\star))^2}{(Q_{x,y}^\gamma(\hat{p}))^2 (Q_{x,y}^\gamma(p^\star))^2}$$

$$\leq \frac{K^2}{\gamma^2 m^2} \min\left\{ \left( 1 - \frac{Q_{x,y}^\gamma(\hat{p})}{Q_{x,y}^\gamma(p^\star)} \right)^2, \left( 1 - \frac{Q_{x,y}^\gamma(p^\star)}{Q_{x,y}^\gamma(\hat{p})} \right) \right\},$$

where we have used the fact that $Q_{x,y}^\gamma(\cdot) \geq \gamma m/K$. Using the fact that $\frac{1}{2}\min\{(1-z)^2, (1-1/z)^2\} \leq -\log z + z - 1$ with $z = Q_{x,y}^\gamma(\hat{p})/Q_{x,y}^\gamma(p^\star)$ (see e.g. [17], Lemma 4) we obtain, after combining the above with Eq. (6) and taking expectations:

$$\|\nabla F(\hat{p}) - \nabla F(p^\star)\|_\infty^2 \leq \frac{2sK^2}{\gamma^2 m^2} \mathbb{E}\left[ \sum_{y \in \mathcal{Y}} r(y) \left\{ -\log \frac{Q_{x,y}^\gamma(\hat{p})}{Q_{x,y}^\gamma(p^\star)} + \frac{Q_{x,y}^\gamma(\hat{p}) - Q_{x,y}^\gamma(p^\star)}{Q_{x,y}^\gamma(p^\star)} \right\} \right]$$

$$= \frac{2sK^2}{\gamma^2 m^2} \mathbb{E}_{z \sim \mathcal{D}'}\left[ f(\hat{p}, z) - f(p^\star, z) - \nabla f(p^\star, z) \cdot (\hat{p} - p^\star) \right]$$

$$= \frac{2sK^2}{\gamma^2 m^2} \left( F(\hat{p}) - F(p^\star) - \nabla F(p^\star) \cdot (\hat{p} - p^\star) \right)$$

$$\leq \frac{2sK^2}{\gamma^2 m^2} \left( F(\hat{p}) - F(p^\star) \right),$$

where in the second inequality we used first-order optimality conditions. $\qquad\square$

*Proof of Theorem 1.* By Lemma 1, Theorem 3 and Lemma 2, phase 1 of Algorithm 1 results in a distribution $\hat{p} \in \Delta_\Pi$ satisfying

$$(1-\gamma)\mathbb{E}_{(x,r) \sim \mathcal{D}}\left[ \sum_{y \in \mathcal{Y}} \frac{r(y)\mathbb{1}\{y \in \pi(x)\}}{Q_{x,y}^\gamma(\hat{p})} \right] \leq 2s \quad \forall \pi \in \Pi,$$

and since $\gamma = 1/2$ we get

$$\mathbb{E}_{(x,r) \sim \mathcal{D}}\left[ \sum_{y \in \pi(x)} \frac{r(y)}{Q_{x,y}^\gamma(\hat{p})} \right] \leq 4s \quad \forall \pi \in \Pi.$$

Fix $\pi \in \Pi$ and for $i \in [N_2]$ define the induced importance-weighted reward estimator by $R_i(\pi) \triangleq \sum_{y \in \pi(x)} \frac{r_i(y) \mathbb{1}\{y \in a_i\}}{Q_{x_i,y}^{\gamma}(\hat{p})}$. This is an unbiased estimator of $r_{\mathcal{D}}(\pi)$, since for a given $y \in \mathcal{Y}$ the probability that $y \in a_i$ equals $Q_{x,y}^{\gamma}(\hat{p})$, and thus

$$\mathbb{E}[R_i(\pi)] = \mathbb{E}\left[\sum_{y \in \pi(x)} r(y)\right] = \mathbb{E}[r \cdot \pi(x)] = r_{\mathcal{D}}(\pi).$$

Moreover, $R_1(\pi), \ldots, R_{N_2}(\pi)$ are i.i.d and exhibit the following variance bound:

$$\begin{aligned}
\mathrm{Var}[R_i(\pi)] &\le \mathbb{E}\left[R_i(\pi)^2\right] \\
&= \mathbb{E}\left[\left(\sum_{y \in \pi(x_i)} \frac{r_i(y)\mathbb{1}\{y \in a_i\}}{Q_{x_i,y}^{\gamma}(\hat{p})}\right)^2\right] \\
&= m^2 \mathbb{E}\left[\left(\frac{1}{m}\sum_{y \in \pi(x_i)} \frac{r_i(y)\mathbb{1}\{y \in a_i\}}{Q_{x_i,y}^{\gamma}(\hat{p})}\right)^2\right] \\
&\le m \mathbb{E}\left[\sum_{y \in \pi(x_i)} \left(\frac{r_i(y)\mathbb{1}\{y \in a_i\}}{Q_{x_i,y}^{\gamma}(\hat{p})}\right)^2\right] \\
&= m \mathbb{E}\left[\sum_{y \in \pi(x_i)} \frac{r_i(y)}{Q_{x_i,y}^{\gamma}(\hat{p})}\right] \le 4sm,
\end{aligned}$$

where in the first inequality we use Jensen's inequality. With the variance bound, we use Bernstein's inequality (e.g. [35], page 86) and the fact that the random variables $R_i(\pi)$ are bounded by $K/(\gamma m)$ to deduce that the following sample complexity suffices to have $r_{\mathcal{D}}(\pi^\star) - r_{\mathcal{D}}(\pi_{\text{out}}) \le \varepsilon$ with probability at least $1 - \delta$:

$$N_1 + N_2 = \Theta\left(\left(\frac{K^9}{m^8} + \frac{K}{m\varepsilon} + \frac{sm}{\varepsilon^2}\right) \log \frac{|\Pi|}{\delta}\right),$$

and the proof is concluded once we note that the second term can be dropped via Young's inequality which shows that the sum of the other two terms dominates the bound. $\qquad\square$

## A.1 Uniform Convergence via Rademacher Complexity

Consider the ERM objective $\widehat{F} : \Delta_\Pi \to \mathbb{R}$ defined as

$$\widehat{F}(p) = \frac{1}{n}\sum_{i=1}^{n} f(p; z_i), \quad p \in \Delta_\Pi,$$

where $f(p; z)$ is defined in Eq. (2) and $z_1, \ldots, z_n$ are sampled i.i.d. from the distribution $\mathcal{D}'$ over $\mathcal{Z}$. Denote by $\mathcal{F} \subseteq \{\mathcal{Z} \to \mathbb{R}\}$ the underlying class of models, that is,

$$\mathcal{F} \triangleq \{z \mapsto f(p; z) \mid p \in \Delta_\Pi\}.$$

For each $z = (x, r, a) \in \mathcal{Z}$ we define the binary matrix $Z \in \{0, 1\}^{K \times |\Pi|}$ by

$$Z_{y,\pi} \triangleq \mathbb{1}\{y \in \pi(x)\}, \quad \forall y \in \mathcal{Y}, \pi \in \Pi.$$

With this notation, we note that the objective can be written as

$$f(p; z = (x, r, a)) = -\frac{K}{m}\sum_{y \in a} r(y) \log\left(\frac{\gamma m}{K} + (1 - \gamma)(Zp)_y\right).$$

Thus, given $y \in \mathcal{Y}$, we define the function $\psi_y(\cdot) : \mathbb{R}_+^K \to \mathbb{R}$ by

$$\psi_y(u) \triangleq -\frac{K}{m} \log\left(\frac{\gamma m}{K} + (1 - \gamma)u_y\right), \quad u \in \mathbb{R}_+^K,$$

so that
$$f(p; z) = \sum_{y \in a} r(y)\psi_y(Zp).$$

Note that for each $y \in \mathcal{Y}$, the function class $\mathcal{L}_y$ defined by
$$\mathcal{L}_y = \{Z_y \mapsto \psi_y(Zp) \mid p \in \Delta_\Pi\},$$

is a class of generalized linear models, as $\psi_y(Zp)$ only depends on the inner product between $p$ and the $y$'th row of the matrix $Z$. Now, given a dataset $S = \{z_1, \ldots, z_n\} \in \mathcal{Z}^n$, the *Rademacher complexity* of $\mathcal{F}$ with respect to $S$ is defined by
$$\mathcal{R}(\mathcal{F} \circ S) \triangleq \frac{1}{n}\mathbb{E}_\sigma\left[\sup_{p \in \Delta_\Pi} \sum_{i=1}^n \sigma_i f(p; z_i)\right],$$

where $\sigma_1, \ldots, \sigma_n$ are i.i.d. Rademacher random variables (that is, $\pm 1$ with probability $\frac{1}{2}$). We make use of the following uniform convergence result over the function class $\mathcal{F}$ in terms of its Rademacher complexity (see e.g. [45]).

**Lemma 4.** *Assume that $|f(\cdot)| \leq B$ for all $f \in \mathcal{F}$. Then with probability at least $1 - \delta$ over $S \sim (\mathcal{D}')^n$, for all $p \in \Delta_\Pi$ it holds that*
$$F(p) - \widehat{F}(p) \leq 2\mathcal{R}(\mathcal{F} \circ S) + 4B\sqrt{\frac{2\log(4/\delta)}{n}}.$$

Note that for our function class of interest $\mathcal{F}$, we have $B \leq \frac{sK}{m}\log\left(\frac{K}{\gamma m}\right)$.

The following key lemma will allow us to obtain a bound on the Rademacher complexity of $\mathcal{F}$ by presenting the functions in the class as linear combinations of $L_\infty$-Lipschitz functions, and using the result given in [20] to relate its Rademacher complexity to that of generalized linear models.

**Lemma 5.** *Let $S \in \mathcal{Z}^n$ where $n \geq \log|\Pi|$. Then,*
$$\mathcal{R}(\mathcal{F} \circ S) \leq O\left(\frac{sK^{2.5}}{m^2}\sqrt{\frac{\log|\Pi|}{n}} \cdot \log^2 n\right).$$

*Proof.* Fix a dataset $S \in \mathcal{Z}^n$. We first write the Rademacher complexity of $\mathcal{F}$ with respect to $S$ as follows:
$$n\mathcal{R}(\mathcal{F} \circ S) = \mathbb{E}_\sigma\left[\sup_{p \in \Delta_\Pi} \sum_{i=1}^n \sigma_i f(p; z_i)\right]$$
$$= \mathbb{E}_\sigma\left[\sup_{p \in \Delta_\Pi} \sum_{i=1}^n \sigma_i \phi_i(Z_i p)\right],$$

where $\phi_1, \ldots, \phi_n : \mathbb{R}_+^K \to \mathbb{R}$ are defined by
$$\phi_i(u) = \sum_{y \in a_i} r_i(y)\psi_y(u), \quad u \in \mathbb{R}^K.$$

Using the $L_\infty$ contraction argument given by Theorem 1 of [20], it holds that
$$\mathbb{E}_\sigma\left[\sup_{p \in \Delta_\Pi} \sum_{i=1}^n \sigma_i \phi_i(Z_i p)\right] \leq O\left(G\sqrt{K}\right) \cdot \max_{y \in \mathcal{Y}} \sup_{S' \in \mathcal{Z}^n} \mathbb{E}_\sigma\left[\sup_{p \in \Delta_\Pi} \sum_{i=1}^n \sigma_i (Z'_{i,y})^\top p\right] \cdot \log^2 n,$$

where $\phi_1, \ldots, \phi_n$ are $G$-Lipschitz with respect to the $L_\infty$-norm. Moreover, the expectation on the right-hand side is the worst-case Rademacher complexity of a linear model that is bounded by 1 in $L_1$-norm over data which is bounded by 1 in $L_\infty$-norm. Using standard Rademacher complexity arguments (e.g. Lemma 26.11 in [45]), this expression is bounded by
$$\max_{y \in \mathcal{Y}} \sup_{S' \in \mathcal{Z}^n} \mathbb{E}_\sigma\left[\sup_{p \in \Delta_\Pi} \sum_{i=1}^n \sigma_i (Z'_{i,y})^\top p\right] \leq \sqrt{2n\log(2|\Pi|)}.$$

It is now left to bound the $L_\infty$-Lipschitz constant of the functions $\phi_1, \ldots, \phi_n$. For all $u, v \in \mathbb{R}_+^K$ and $i \in [n]$, we have

$$
\begin{aligned}
|\phi_i(u) - \phi_i(v)| &= \left| \sum_{y \in a_i} r_i(y)\big(\psi_y(u) - \psi_y(v)\big) \right| \\
&\leq \sum_{y \in a_i} r_i(y)\big|\psi_y(u) - \psi_y(v)\big| \\
&\leq s \cdot \max_{y \in \mathcal{Y}} \big|\psi_y(u) - \psi_y(v)\big|,
\end{aligned}
$$

where we used the fact that $\|r_i\|_1 \leq s$. Now, for each $y \in \mathcal{Y}$, it holds that

$$
\|\nabla \psi_y(u)\|_1 = \left| \frac{\partial}{\partial u_y} \psi_y(u) \right| = \frac{K}{m} \cdot \frac{1 - \gamma}{\frac{\gamma m}{K} + (1 - \gamma)u_y} \leq \frac{K^2}{\gamma m^2}.
$$

Therefore, using Hölder's inequality,

$$
|\phi_i(u) - \phi_i(v)| \leq s \cdot \frac{K^2}{\gamma m^2} \cdot \|u - v\|_\infty,
$$

and we deduce that $G \leq \frac{sK^2}{\gamma m^2}$, which concludes the proof. $\qquad\square$

We can now use Lemma 4 and Lemma 5 to prove Lemma 2.

*Proof of Lemma 2.* Let $\hat{p} \in \Delta_\Pi$ be the output of Algorithm 2 when run on $\widehat{F}(\cdot)$ using the dataset $S$ of $n$ i.i.d. samples from $\mathcal{D}'$ and assume that $\widehat{F}(\hat{p}) - \widehat{F}(p) \leq \mu/3$ for all $p \in \Delta_\Pi$. Let $p_\star = \operatorname{argmin}_{p \in \Delta_\Pi} F(p)$. Then with probability at least $1 - \delta$,

$$
\begin{aligned}
F(\hat{p}) - F(p) &= F(\hat{p}) - \widehat{F}(\hat{p}) + \widehat{F}(\hat{p}) - F(p) \\
&\leq F(\hat{p}) - \widehat{F}(\hat{p}) + \widehat{F}(p) - F(p) + \frac{\mu}{2} \\
&\leq F(\hat{p}) - \widehat{F}(\hat{p}) + \mu/3 + B\sqrt{\frac{\log(2/\delta)}{2n}},
\end{aligned}
$$

where the first inequality uses the fact that $\hat{p}$ is a $(\mu/3)$-approximate minimizer of $\widehat{F}(\cdot)$ and the second inequality follows from Hoeffding's inequality with $B = \frac{sK}{m} \log(K/(\gamma m))$ being a bound on both the empirical and the expected function values. We now use Lemma 4 to further bound the suboptimality of $\hat{p}$ by

$$
F(\hat{p}) - F(p) = 2\mathcal{R}(\mathcal{F} \circ S) + 5B\sqrt{\frac{2\log(4/\delta)}{n}} + \frac{\mu}{3}.
$$

The proof is now concluded using Lemma 5 once we note that if $n = \widetilde{\Theta}\left( \frac{s^2 K^5}{m^4 \mu^2} \log(|\Pi|/\delta) \right)$ then the above is bounded by $\mu$. $\qquad\square$

## A.2  The Frank-Wolfe Algorithm

We now present the details of the Frank-Wolfe optimization algorithm used to obtain an approximate minimizer of the objective defined in Eq. (1). The algorithm operates under the classical convex optimization model:

$$
\text{minimize} \quad F(w), \quad w \in \mathcal{W},
$$

where $\mathcal{W} \subseteq \mathbb{R}^d$ is a convex domain. We assume that the algorithm has access to the full gradients of the objective, that is, it has access to $\nabla F(w)$ for each $w \in \mathcal{W}$. We assume the objective satisfies the following $L_1$-smoothness condition:

For all $u, w \in \mathcal{W}$ and it holds that $F(w) \leq F(u) + \nabla F(u)^\top (w - u) + \frac{\beta}{2}\|w - u\|_1^2$.

Applying the generic result of given in Theorem 1 of [29] to the objective defined in Eq. (1), we obtain the following result stating the rate of convergence and oracle complexity of Algorithm 2.

---

**Algorithm 2** Frank-Wolfe

---

Parameters: Objective $F : \Delta_\Pi \to \mathbb{R}$, step sizes $\{\eta_t\}_t$.
Initialize $p_1 \in \Delta_\Pi$.
**for** $t = 1, 2, \ldots$ **do**
    Let $g_t = \nabla F(p_t)$.
    Compute $q_t = \mathsf{LOO}_\Pi(g_t)$;
    Update $p_{t+1} = (1 - \eta_t)p_t + \eta_t q_t$;
**end for**

---

**Theorem 3.** *Let* $p^\star = \operatorname{argmin}_{p \in \Delta_\Pi} \widehat{F}(p)$. *Then Algorithm 2 with step sizes* $\eta_t = 2/(2+t)$ *outputs* $\hat{p} \in \Delta_\Pi$ *with* $\widehat{F}(\hat{p}) - \widehat{F}(p^\star) \leq \mu$ *within* $T$ *iterations, where*

$$T = O\left( \frac{sK^3}{\gamma^2 m^3 \mu} \right).$$

*In particular, Algorithm 2 makes at most* $O\big(sK^3/(\gamma^2 m^3 \mu)\big)$ *calls to* $\mathsf{LOO}$.

*Proof.* Using Theorem 1 of [29], it suffices to prove that the objective $\widehat{F}(\cdot)$ defined in Eq. (1) satisfies the following $L_1$-smoothness property:

$$\|\nabla \widehat{F}(p) - \nabla \widehat{F}(q)\|_\infty \leq \frac{sK^3}{\gamma^2 m^3} \|p - q\|_1.$$

Indeed, observe that for all $\pi \in \Pi$:

$$\left| \left( \nabla \widehat{F}(p) \right)_\pi - \left( \nabla \widehat{F}(q) \right)_\pi \right| \leq \frac{1}{N_1} \sum_{i=1}^{N_1} \left| (\nabla f(p; z_i))_\pi - (\nabla f(q; z_i))_\pi \right|$$

$$\leq \sup_{z=(x,r,a) \in \mathcal{Z}} \frac{K}{m} \sum_{y \in a} r(y) \left| \frac{1}{Q^\gamma_{x,y}(p)} - \frac{1}{Q^\gamma_{x,y}(q)} \right|$$

$$= \sup_{z=(x,r,a) \in \mathcal{Z}} (1 - \gamma) \frac{K}{m} \sum_{y \in a} r(y) \left| \frac{Q^\gamma_{x,y}(p) - Q^\gamma_{x,y}(p)}{Q^\gamma_{x,y}(p) Q^\gamma_{x,y}(q)} \right|$$

$$\leq \frac{sK^3}{\gamma^2 m^3} \|p - q\|_1,$$

where we have used the fact that $\|r\|_1 \leq s$. $\qquad\square$

The following lemma asserts that each call to the linear optimization oracle in Algorithm 2 can in fact be implemented by a call to the ERM oracle $\mathsf{ERM}_\Pi$.

**Lemma 6.** *In Algorithm 2 with the objective defined in Eq.* (1)*, each call to* $\mathsf{LOO}_\Pi$ *can be implemented via a single call to* $\mathsf{ERM}_\Pi$.

*Proof.* For each $t = 1, 2, \ldots$ it holds that

$$\mathsf{LOO}_\Pi(g_t) = \operatorname*{argmin}_{p \in \Delta_\Pi} \left\{ \nabla \widehat{F}(p_t)^\top p \right\}$$

$$= \operatorname*{argmin}_{p \in \Delta_\Pi} \left\{ \frac{1}{N_1} \sum_{i=1}^{N_1} \nabla f(p_t; z_i)^\top p \right\}$$

$$= \operatorname*{argmin}_{\pi \in \Pi} \left\{ \frac{1}{N_1} \sum_{i=1}^{N_1} (\nabla f(p_t; z_i))_\pi \right\}$$

$$= \operatorname*{argmax}_{\pi \in \Pi} \left\{ \sum_{i=1}^{N_1} \sum_{y \in \pi(x_i)} \hat{r}_i(y) \right\},$$

where $\hat{r}_i(y) = \frac{1}{N_1}(1 - \gamma)\frac{K}{m}\frac{\mathbb{1}\{y \in a_i\}r_i(y)}{Q^\gamma_{x_i,y}(p_t)}$ by Eq. (4). $\qquad\square$

## A.3 Improved Rate for the Single-Label Setting

We now present a version of Algorithm 1 specialized for the single-label classification setting, and show that it enjoys a sample complexity guarantee whose dependence on $K$ is better than that obtained by [17].

---

**Algorithm 3** Bandit PAC for Single Label Classification

---

Parameters: $N_1, N_2, \gamma \in (0, \frac{1}{2}], T \in \mathbb{N}$.

**Phase 1:**

Initialize $S = \emptyset$.

**for** $i = 1, \ldots, N_1$ **do**

    Environment generates $(x_i, y_i) \sim \mathcal{D}$, algorithm receives $x_i$.

    Predict a uniformly random $\hat{y}_i \in \mathcal{Y}$ and receive feedback $\mathbb{1}\{\hat{y}_i = y_i\}$.

    If $\hat{y}_i = y_i$, update $S \leftarrow S \cup \{(x_i, y_i)\}$.

**end for**

Initialize $p_1 \in \Delta_{\mathcal{H}}$ to be a delta distribution.

**for** $t = 1, \ldots, T$ **do**

    Let $q_t \in \Delta_{\mathcal{H}}$ be the delta distribution on $h_t = \mathsf{ERM}_\Pi\left(\{(x_i, \hat{r}_i)\}_{i=1}^{|S|}\right)$, where

$$\hat{r}_i(y) = \frac{1}{|S|}(1 - \gamma)\frac{\mathbb{1}\{\hat{y}_i = y_i = y\}}{Q^\gamma_{x_i, y}(p_t)}, \quad \forall i \in [|S|], y \in \mathcal{Y}.$$

    Update $p_{t+1} = (1 - \eta_t)p_t + \eta_t q_t$ where $\eta_t = 2/(2 + t)$.

**end for**

Let $\hat{p} = p_{T+1}$.

**Phase 2:**

**for** $i = 1, \ldots, N_2$ **do**

    Environment generates $(x_i, y_i) \sim \mathcal{D}$, algorithm receives $x_i$.

    With prob. $\gamma$ pick $\hat{y}_i \in \mathcal{Y}$ uniformly at random; otherwise sample $h_i \sim \hat{p}$ and set $\hat{y}_i = h_i(x_i)$.

    Predict $\hat{y}_i$ and receive feedback $\mathbb{1}\{\hat{y}_i = y_i\}$.

**end for**

Return:

$$h_{\text{out}} = \mathsf{ERM}_{\mathcal{H}}\left(\{(x_i, \hat{r}_i)\}_{i=1}^{N_2}\right), \quad \text{where} \quad \hat{r}_i(y) = \frac{\mathbb{1}\{\hat{y}_i = y_i = y\}}{Q^\gamma_{x_i, y}(\hat{p})} \quad \forall y \in \mathcal{Y}.$$

---

At a high level, the difference between the specialized version given in Algorithm 3 and the more general approach (Algorithm 1) lies in the fact that in the single label setting, we are able to collect a fully-labeled dataset by predicting random labels, and once every $\approx K$ rounds we are guaranteed to add a sample to our dataset with high probability, as is the approach of [17]. This removes the need to estimate the objective given in Eq. (3) via importance sampling, which would result in a scaling factor of $K$ in the objective and thus in an additional $K^2$ factor in the total sample complexity due to the Rademacher complexity arguments. Instead, collecting the dataset $S$ results in one extra factor $K$ in the total sample complexity. The improvement of the sample complexity bound over that of [18] arises from the usage of the empirical FW optimization algorithm instead of the stochastic variant. Indeed, in our approach, the sample complexity depends on the Lipschitz constant of the objective which scales with $K$, and not on its smoothness parameter which scales with $K^2$. The result is stated formally in the following.

**Theorem 4.** *Consider the single-label bandit multiclass classification setting. If we set $\gamma = \frac{1}{2}$, $N_1 = \Theta(K^7 \log(|\mathcal{H}|/\delta))$, $N_2 = \Theta\left((K/\varepsilon + 1/\varepsilon^2)\log(|\mathcal{H}|/\delta)\right)$ and $T = \Theta(K^4)$ in Algorithm 3, then with probability at least $1 - \delta$ Algorithm 3 outputs $h_{\text{out}} \in \mathcal{H}$ with $r_{\mathcal{D}}(h^\star) - r_{\mathcal{D}}(h_{\text{out}}) \leq \varepsilon$ using a sample complexity of*

$$N_1 + N_2 = O\left(\left(K^7 + \frac{1}{\varepsilon^2}\right)\log\frac{|\mathcal{H}|}{\delta}\right).$$

*Furthermore, Algorithm 3 makes a total of $T + 1 = O(K^4)$ calls to $\mathsf{ERM}_{\mathcal{H}}$.*

*Proof.* The analysis of the second phase is identical to that of the more general setting of combinatorial semi-bandits with $s$-sparse rewards (see the proof of Theorem 1), where in this case we have $s = m = 1$. Now, note that the first phase of Algorithm 3 is an implementation of the FW algorithm on the following objective:

$$\text{minimize} \quad \widehat{F}(p) = \frac{1}{|S|} \sum_{(x,y) \in S} \left[ -\log(Q_{x,y}^{\gamma}(p)) \right], \quad p \in \Delta_{\mathcal{H}}.$$

Thus, it suffices to show that in the first phase, $O\big(K^7 \log(|\mathcal{H}|/\delta)\big)$ samples suffice in for uniform converges of $\widehat{F}(\cdot)$ with rate $\mu = 1/8K^2$. Indeed, a straightforward concentration argument shows that with probability at least $1 - \delta$, the number of samples in $S$ after the data collection process is at least

$$|S| = \Omega\left( K^6 \log \frac{|\mathcal{H}|}{\delta} \right).$$

Now, note that the realized objectives optimized in the first phase come from a generalized linear model of the form

$$\mathcal{F} = \left\{ z \mapsto -\log(\gamma/K + (1 - \gamma)z^{\top}p) \mid p \in \Delta_{\mathcal{H}} \right\}.$$

Since the function $\psi : \mathbb{R} \to \mathbb{R}$ defined by $\psi(x) = -\log(\gamma/K + (1 - \gamma)x)$ is $G$-Lipschitz with $G = K/\gamma$, a simple contraction argument (see e.g. Lemma 26.9 in [45]) together with a bound on the Rademacher complexity of $L_1/L_\infty$-bounded linear models (see e.g. Lemma 26.11 in [45]), shows that the Rademacher complexity of $\mathcal{F}$ with respect to $S$ is bounded by

$$\mathcal{R}(\mathcal{F} \circ S) \le G \cdot \sqrt{\frac{2\log(2|\mathcal{H}|)}{|S|}} = \frac{K}{\gamma} \sqrt{\frac{2\log(2|\mathcal{H}|)}{|S|}}.$$

Using Lemma 4, as in the proof of Lemma 2, we deduce that with probability at least $1 - \delta$ it holds that

$$F(\hat{p}) - F(p_{\star}) \le \mu,$$

where $\mu = 1/8K^2$. The bound on the number of iterations of FW follows from the fact that the objective is $O(K^2)$-smooth with respect to the $L_1$ norm, and similar arguments as those in the proof of Theorem 3. $\qquad \square$

### A.4 Lower Bound for Sparse Combinatorial Semi-Bandits

We now provide a formal proof of the sample complexity lower bound given in Theorem 2 (and more formally restated below) for combinatorial semi-bandits with $s$-sparse rewards. We construct hard instances for the problem denoted by $\mathcal{I}_S$ for all subsets $S \subseteq \mathcal{Y}$ of size $m$. In $\mathcal{I}_S$, the reward distribution is constructed as follows: The rewards of all actions $y \in S$ are Bernoulli random variables with parameter $\frac{s}{2K} + \frac{\varepsilon}{m}$, and the rewards of all actions $y \in \mathcal{Y} \setminus S$ are Bernoulli random variables with parameter $\frac{s}{2K} - \frac{\varepsilon}{K-m}$. The rewards are sampled in an i.i.d. manner between actions, and it is easily seen that the sum of expected rewards of actions is exactly $\frac{s}{2}$. Using standard binomial concentration arguments, we will show that with high probability all realized reward vectors are $s$-sparse, and so conditioning on the event in which all reward realizations are $s$-sparse will not substantially affect the lower bound. We also limit our analysis to deterministic algorithms, which suffices in general due to Yao's principle as the instances we construct do not depend on the decisions of the algorithm. The formal lower bound is established in the following theorem:

**Theorem 5** (restatement of Theorem 2). *Let* Alg *be a deterministic combinatorial semi-bandit algorithm over action set $\mathcal{Y}$ of size $K$ with combinatorial actions being subsets of $\mathcal{Y}$ of size $m \le K/12$. Then for sufficiently small $\varepsilon > 0$, there exists an instance $\mathcal{I}_S$ for which if we run* Alg *for $T \le \frac{sm}{3072\varepsilon^2}$ rounds over rewards sampled according to $\mathcal{I}_S$, then* Alg *outputs an $\varepsilon/2$-suboptimal subset with probability at least $1/4$.*

*Proof.* For the analysis, we define an additional problem instance denoted by $\mathcal{I}_0$ in which the expected reward of all actions is $\frac{s}{2K} - \frac{\varepsilon}{K-m}$. For a given $S \subseteq \mathcal{Y}$ of size $m$, we denote by $\mathbb{P}_S$ and $\mathbb{P}_0$ the probability distributions over length $T$ sequences $(a_1, r_1, \dots, a_T, r_T)$ where $a_t$ is the subset produced by Alg on round $t$, conditioned on the instance $\mathcal{I}_S$ or $\mathcal{I}_0$, respectively. We also denote the history up

to round $t$ by $h_t = (a_1, r_1, \ldots, a_{t-1}, r_{t-1})$, and note that since Alg is assumed to be deterministic, $a_t$ is deterministically chosen conditioned on $h_t$. For an action $y \in \mathcal{Y}$ denote by $N_y$ the number of times Alg chooses a subset containing $y$ during the $T$ rounds.

We claim that there is some subset $\mathcal{Y}' \subseteq \mathcal{Y}$ of size at least $K/3$ such that for all $y \in \mathcal{Y}'$ it holds that

$$\mathbb{E}_0[N_y] \le \frac{3T}{K} \text{ and } \mathbb{P}_0[y \in a_T] \le \frac{3m}{K}.$$

Indeed, by the fact that each action may be chosen at most $T$ times, for at least $(2/3)K$ actions $y \in \mathcal{Y}$ it holds that

$$\mathbb{E}_0[N_y] \le \frac{3T}{K}.$$

Similarly, since at each round Alg picks $m$ actions, for at least $(2/3)K$ actions $y \in \mathcal{Y}$ it holds that

$$\mathbb{P}_0[y \in a_T] \le \frac{3m}{K}.$$

Therefore, both conclusions hold for the intersection of the two subsets of actions, denoted by $\mathcal{Y}'$, which must be of size at least $K/3$. Let $\mathcal{S} \subseteq \mathcal{Y}'$ be some subset of $\mathcal{Y}'$ of size $m$. We now show that the theorem holds with the instance $\mathcal{I}_\mathcal{S}$.

Now, by construction of $\mathcal{I}_\mathcal{S}$, in order to prove the theorem it suffices to upper bound the probability that at least half of the actions in $\mathcal{S}$ belong to $a_T$ by $3/4$. By Markov's inequality, it then suffices to prove that the expected number of arms of $\mathcal{S}$ which belong to $a_T$ is at most $3m/8$. Fix $y \in \mathcal{S}$. By Pinsker's inequality and the chain rule for KL-divergence, it holds that

$$2(\mathbb{P}_0[y \in a_T] - \mathbb{P}_\mathcal{S}[y \in a_T])^2 \le \mathsf{KL}(\mathbb{P}_0 \mid \mathbb{P}_\mathcal{S})$$

$$= \sum_{t=1}^{T} \mathsf{KL}(\mathbb{P}_0[r_t \mid h_t] \mid \mathbb{P}_\mathcal{S}[r_t \mid h_t])$$

$$= \sum_{t=1}^{T} \sum_{\mathcal{S}' \subseteq \mathcal{Y}, |\mathcal{S}'|=m} \mathbb{P}_0[a_t = \mathcal{S}'] \sum_{y' \in \mathcal{S}' \cap \mathcal{S}} \mathsf{KL}(\mathbb{P}_0[r_t(y') \mid h_t] \mid \mathbb{P}_\mathcal{S}[r_t(y') \mid h_t]),$$

where $\mathsf{KL}(p \mid q) = \sum_z p(z) \log \frac{p(z)}{q(z)}$ is the KL-divergence between distributions. Now, note that the KL terms in the last expression are all equal to the KL divergence between Bernoulli random variables with biases $\frac{s}{2K} - \frac{\varepsilon}{K-m}$ and $\frac{s}{2K} + \frac{\varepsilon}{m}$, respectively, and thus can be bounded for sufficiently small $\varepsilon$ by

$$\mathsf{KL}\left(\mathrm{Ber}\left(\frac{s}{2K} - \frac{\varepsilon}{K-m}\right) \mid \mathrm{Ber}\left(\frac{s}{2K} + \frac{\varepsilon}{m}\right)\right) \le \frac{(2\varepsilon/m)^2}{\left(\frac{s}{2K} - \frac{\varepsilon}{K-m}\right)\left(1 - \frac{s}{2K} + \frac{\varepsilon}{K-m}\right)} \le \frac{32\varepsilon^2 K}{m^2 s}.$$

Therefore, the above is further bounded by

$$2(\mathbb{P}_0[y \in a_T] - \mathbb{P}_\mathcal{S}[y \in a_T])^2 \le \sum_{t=1}^{T} \sum_{\mathcal{S}' \subseteq \mathcal{Y}, |\mathcal{S}'|=m} \mathbb{P}_0[a_t = \mathcal{S}'] |\mathcal{S}' \cap \mathcal{S}| \cdot \frac{32\varepsilon^2 K}{m^2 s}$$

$$= \frac{32\varepsilon^2 K}{m^2 s} \sum_{y' \in \mathcal{S}} \mathbb{E}_0[N_{y'}]$$

$$\le \frac{96\varepsilon^2 T}{ms}$$

$$\le \frac{1}{32},$$

where we used the fact that $\mathbb{E}_0[N_{y'}] \le 3T/K$ for each $y' \in \mathcal{S}$ and the choice of $T$. Simplifying the above inequality and using the fact that $\mathbb{P}_0[y \in a_T] \le 3m/K$, we have

$$\mathbb{P}_\mathcal{S}[y \in a_T] \le \frac{1}{8} + \frac{3m}{K} \le \frac{1}{8} + \frac{1}{4} = \frac{3}{8},$$

where we used the fact that $m \leq K/12$. Summing over $\mathcal{S}$, we obtain

$$\sum_{y \in \mathcal{S}} \mathbb{P}_{\mathcal{S}}[y \in a_T] \leq \frac{3m}{8},$$

which concludes the proof. $\qquad\square$

In order to obtain a valid lower bound for instances in which the reward realizations are $s$-sparse, we use a multiplicative Chernoff bound together with a union bound over $[T]$ to deduce that for every instance $\mathcal{I}_{\mathcal{S}}$:

$$\mathbb{P}_{\mathcal{S}}[\exists t \in [T] \text{ s.t. } \|r_t\|_1 > s] \leq Te^{-s/4} \leq \frac{1}{2},$$

where the last inequality holds for $s \geq 4\log(2T)$. Note that conditioned on the event that $\|r_t\|_1 \leq s$ for all $t$, the rewards are still i.i.d. across rounds, and thus their conditional distribution defines an $s$-sparse instance on which Alg outputs an $\varepsilon/2$-suboptimal subset with probability at least $\frac{1}{2}$.

# B  Online Regret Minimization Setting

In this section we present a regret minimization algorithm for CCSB with rewards that satisfy $\|r_t\|_2^2 \leq s$ which is a slightly relaxed version of the assumption $\|r_t\|_1 \leq s$. Instead of rewards in $[0, 1]$, it will be convenient for us to instead consider losses, so we introduce the following negative loss vectors:

$$\ell_t(y) = -r_t(y) \quad \forall t \in [T], y \in \mathcal{Y}.$$

It is obvious that the losses satisfy $\|\ell_t\|_2^2 \leq s$ and that the transformation from rewards to losses does not affect the regret. A natural approach for regret minimization in CCSB would be to use the EXP4 algorithm [5] over the policy class $\Pi$, which amounts to Follow-the-regularized-leader (FTRL) using *negative entropy* regularization, defined as

$$H(p) \triangleq \sum_{i=1}^{|\Pi|} p_i \log p_i, \quad p \in \Delta_\Pi. \tag{7}$$

However, as been observed by [18], since the loss vectors are negative, this approach alone would not suffice to achieve a regret bound that is adaptive to the sparsity level $s$, and would instead yield a regret bound of the form $O(\sqrt{mKT \log |\Pi|})$. In order to adapt to sparsity, they introduce an additional *log-barrier* regularization, defined as

$$\Phi(p) \triangleq \sum_{i=1}^{|\Pi|} \log p_i, \quad p \in \Delta_\Pi. \tag{8}$$

We adopt this approach and generalize it to the combinatorial setup with Algorithm 4.

We prove the following result for Algorithm 4:

**Theorem 6.** *Assume that the loss vectors $\ell_t$ satisfy $\|\ell_t\|_2^2 \leq s$ for all $t \in [T]$. Then Algorithm 4 with $\nu = \frac{1}{16}$ and $\eta = \sqrt{\log(|\Pi|)/(msT)}$ attains the following expected regret bound w.r.t. a finite policy class $\Pi$:*

$$\mathbb{E}[\mathcal{R}_T] \leq O\left(|\Pi| \log(|\Pi|T) + \sqrt{smT \log |\Pi|}\right).$$

We remark that the linear dependence on $|\Pi|$ is essentially tight if we require the leading term to depend on $s$ rather than $K$, as shown in [18] for the single-label setting.

Theorem 6 primarily stems from the following the following result which is a consequence of a generic analysis of FTRL (see e.g. [26], [41] (Lemma 7.14)):

**Algorithm 4** EXP4-COMB-SPARSE

---

Parameters: $m, K, T, s$, finite policy class $\Pi \subseteq \{X \to \mathcal{A}\}$, step sizes $\eta > 0, 0 < \nu \leq 1$.
Initialize $p_1 \in \Delta_\Pi$ as the uniform distribution over $\Pi$.
**for** $t = 1, 2, \ldots, T$ **do**
    Obtain $x_t \in X$.
    Sample $\pi_t \sim p_t$ and let $a_t = \pi_t(x_t) \in \mathcal{A}$.
    Observe $\{\ell_t(y) \mid y \in a_t\}$ and construct importance-weighted loss estimators for policies in $\Pi$
    via

$$\hat{c}_t(i) = \sum_{y \in \pi_i(x_t)} \frac{\ell_t(y)\mathbf{1}\{y \in a_t\}}{Q_t(y)} \quad \forall i \in [|\Pi|],$$

    where $Q_t(y) = \sum_{i=1}^{|\Pi|} p_t(i)\mathbb{1}\{y \in \pi_i(x_t)\}$ is the probability that $y$ belongs to $a_t$ when sampling a
    policy using $p_t$.
    Update $p_t$ via

$$p_{t+1} = \underset{p \in \Delta_\Pi}{\arg\min}\left\{p \cdot \sum_{\tau=1}^{t} \hat{c}_\tau + \frac{1}{\eta}H(p) + \frac{1}{\nu}\Phi(p)\right\},$$

    where $H(\cdot)$ and $\Phi(\cdot)$ are defined in Eq. (7) and Eq. (8) respectively.
**end for**

---

**Lemma 7.** *For all $p^\star \in \Delta_\Pi$, the following regret bound holds for Algorithm 4:*

$$\sum_{t=1}^{T} \hat{c}_t \cdot (p_t - p^\star) \leq R(p^\star) - R(p_1) + \frac{\eta}{2}\sum_{t=1}^{T}\sum_{i=1}^{|\Pi|} \tilde{p}_t(i)\hat{c}_t(i)^2,$$

*where $R(p) \triangleq \frac{1}{\eta}H(p) + \frac{1}{\nu}\Phi(p)$, and $\tilde{p}_t \in [p_t, p_{t+1}]$ is some point which lies on the line segment connecting $p_t$ and $p_{t+1}$.*

In order to relate $\tilde{p}_t$ given in the bound to $p_t$, we use the following result which is where we make use of the properties of the log-barrier regularization $\Phi(\cdot)$.

**Lemma 8.** *Assuming that $\nu \leq \frac{1}{16}$, it holds that $p_{t+1}(i) \leq \frac{1}{8\nu}p_t(i)$ for all $i \in [|\Pi|]$.*

[18] prove this claim for the single-label setting, for completeness we include the proof for the multilabel setting. The proof requires the following preliminary definition:

**Local and dual norms.** Given a strictly convex twice-differentiable function $F : \mathcal{W} \to \mathbb{R}$ where $\mathcal{W} \subseteq \mathbb{R}^d$ is a convex domain, we define the *local norm* of a vector $g \in \mathcal{W}$ about a point $z \in \mathcal{W}$ with respect to $F$ by

$$\|g\|_{F,z} \triangleq \sqrt{g^\top \nabla^2 F(z)g},$$

and its *dual norm* by

$$\|g\|_{F,z}^* \triangleq \sqrt{g^\top \nabla^2 F(z)^{-1}g}.$$

*Proof of Lemma 8.* Fix $t \in [T]$ and define $F_t : \Delta_\Pi \to \mathbb{R}$ by

$$F_t(p) \triangleq \sum_{\tau=1}^{t-1} \hat{c}_\tau \cdot p + R(p),$$

where $R(p) \triangleq H_\eta(p) + \Phi_\nu(p) \triangleq \frac{1}{\eta}H(p) + \frac{1}{\nu}\Phi(p)$. That is, $F_t$ is the function minimized by $p_t$ at round $t$ of Algorithm 4. Now, by the form of the Hessian of $\Phi(\cdot)$, for all $p, p', q \in \Delta_\Pi$ it holds that

$$\|p - p'\|_{\Phi_\nu,q}^2 = \frac{1}{\nu}\sum_{i=1}^{|\Pi|} \frac{(p(i) - p'(i))^2}{q(i)^2},$$

and thus it suffices to prove that $\|p_{t+1} - p_t\|^2_{\Phi_\nu, p_t} \leq \frac{c^2}{\nu}$ where $c \triangleq \frac{1}{8\nu} - 1 \geq 1$, since in this case for all $i \in [|\Pi|]$ we will have

$$(p_{t+1}(i) - p_t(i))^2 \leq \left(\frac{1}{8\nu} - 1\right)^2 p_t(i)^2,$$

implying the result. Note that a sufficient condition for the above is that for all $q \in \Delta_\Pi$ for which $\|q - p_t\|_{\Phi_\nu, p_t} = \frac{c^2}{\nu}$ it holds that $F_{t+1}(q) \geq F_{t+1}(p_t)$. Indeed, in this case, if we define the convex set $\mathcal{E} \triangleq \left\{ p \in \Delta_\Pi \mid \|p - p_t\|_{\Phi_\nu, p_t} \leq \frac{c^2}{\nu} \right\}$ (which in particular contains $p_t$), since $F_{t+1}$ is strictly convex, the condition that $F_{t+1}(q) \geq F_{t+1}(p_t)$ for all $q$ on the relative boundary of $\mathcal{E}$ implies that its minimizer, $p_{t+1}$, must belong to $\mathcal{E}$. With that in mind, fix $q \in \Delta_\Pi$ with $\|q - p_t\|^2_{\Phi_\nu, p_t} = \frac{c^2}{\nu}$. Using a second-order Taylor approximation of $F_{t+1}$ around $p_t$, we have

$$F_{t+1}(q) \geq F_{t+1}(p_t) + \nabla F_{t+1}(p_t)^\top (q - p_t) + \frac{1}{2}\|q - p_t\|^2_{R, p},$$

where $p$ is a point on the line segment connecting $p_t$ and $q$. Using the definition of $F_{t+1}$ and the fact that $\nabla^2 R(\cdot) \succeq \nabla^2 \Phi_\nu(\cdot)$ since $H_\eta$ is convex, we get

$$F_{t+1}(q) \geq F_{t+1}(p_t) + \nabla F_t(p_t)^\top (q - p_t) + \hat{c}_t^\top (q - p_t) + \frac{1}{2}\|q - p_t\|^2_{\Phi_\nu, p},$$

and using first-order convex optimality conditions for $p_t$ as a minimizer of $F_t$, we obtain

$$F_{t+1}(q) \geq F_{t+1}(p_t) + \hat{c}_t^\top (q - p_t) + \frac{1}{2}\|q - p_t\|^2_{\Phi_\nu, p}.$$

Starting with the local norm term, we note that since $\|q - p_t\|^2_{\Phi_\nu, p_t} \leq \frac{c^2}{\nu}$ it holds that $q(i) \leq \frac{1}{8\nu} p_t(i)$ for all $i \in [|\Pi|]$, and since $p$ lies on the segment connecting $q$ and $p_t$, the same holds for $p$ instead of $q$. Therefore,

$$
\begin{aligned}
\|q - p_t\|^2_{\Phi_\nu, p} &= \frac{1}{\nu} \sum_{i=1}^{|\Pi|} \frac{(q(i) - p_t(i))^2}{p(i)^2} \\
&\geq 64\nu \sum_{i=1}^{|\Pi|} \frac{(q(i) - p_t(i))^2}{p_t(i)^2} \\
&= 64\nu^2 \|q - p_t\|^2_{\Phi_\nu, p_t} \\
&= 64\nu c^2.
\end{aligned}
$$

Thus, to conclude the proof, we need to show that $\hat{c}_t^\top (q - p_t) \geq -32\nu c^2$. Indeed, since the losses are non-positive, we have

$$
\begin{aligned}
\hat{c}_t^\top (q - p_t) &= \frac{\ell_t(a_t)}{Q_t(a_t)} \sum_{i=1}^{|\Pi|} (q(i) - p_t(i)) \mathbb{1}\{\pi_i(x_t) = a_t\} \\
&\geq \frac{\ell_t(a_t)}{Q_t(a_t)} \sum_{i=1}^{|\Pi|} q(i) \mathbb{1}\{\pi_i(x_t) = a_t\}.
\end{aligned}
$$

Using the fact that $q(i) \leq \frac{1}{8\nu} p_t(i)$ we further lower bound this term as

$$
\begin{aligned}
\hat{c}_t^\top (q - p_t) &\geq \frac{1}{8\nu} \frac{\ell_t(a_t)}{Q_t(a_t)} \sum_{i=1}^{|\Pi|} p_t(i) \mathbb{1}\{\pi_i(x_t) = a_t\} \\
&= \frac{1}{8\nu} \ell_t(a_t) \\
&\geq -\frac{1}{8\nu},
\end{aligned}
$$

where we used the fact that $\ell_t(\cdot) \in [-1, 0]$. The proof is concluded once we note that $32\nu c^2 \geq \frac{1}{8\nu}$ if and only if $c^2 \geq 256\nu^2$, which clearly holds since $256\nu^2 \leq 1 \leq c^2$. $\qquad \square$

With Lemma 7 and Lemma 8 in hand, we can prove the following result:

**Theorem 7.** *Algorithm 4 with $v \leq \frac{1}{16}$ and $\eta > 0$ attains the following expected regret bound:*

$$\mathbb{E}[\mathcal{R}_T] \leq 1 + \frac{|\Pi| \log(|\Pi|T)}{v} + \frac{\log |\Pi|}{\eta} + \eta \mathbb{E}\left[\sum_{t=1}^{T} \sum_{i=1}^{|\Pi|} p_t(i) \hat{c}_t(i)^2\right],$$

*Proof of Theorem 7.* Fix $p^\star \in \Delta_\Pi$ and for $\gamma = \frac{1}{|\Pi|T}$ let $p_\gamma^\star(i) \triangleq (1 - |\Pi|\gamma)p^\star(i) + \gamma$ for all $i \in [|\Pi|]$. In addition, let $c_t \in [-1, 0]^{|\Pi|}$ defined by $c_t(i) = \ell_t(\pi_i(x_t))$. We have,

$$\sum_{t=1}^{T} c_t \cdot (p_t - p^\star) = \sum_{t=1}^{T} c_t \cdot \left(p_t - p_\gamma^\star\right) + \sum_{t=1}^{T} c_t \cdot \left(p_\gamma^\star - p^\star\right)$$

$$= \sum_{t=1}^{T} c_t \cdot \left(p_t - p_\gamma^\star\right) + \sum_{t=1}^{T} \sum_{i=1}^{|\Pi|} c_t(i)\left(\gamma - |\Pi|\gamma p^\star(i)\right)$$

$$\leq \sum_{t=1}^{T} c_t \cdot \left(p_t - p_\gamma^\star\right) + \gamma|\Pi|T$$

$$= \sum_{t=1}^{T} c_t \cdot \left(p_t - p_\gamma^\star\right) + 1$$

where we have used the fact that $c_t(i) \in [-1, 0]$. Taking expectations and using Lemma 7 and Lemma 8 together with the fact that $\mathbb{E}_t[\hat{c}_t] = c_t$, we obtain

$$\mathbb{E}[\mathcal{R}_T] \leq 1 + \mathbb{E}\left[\sum_{t=1}^{T} \mathbb{E}_t[\hat{c}_t] \cdot \left(p_t - p_\gamma^\star\right)\right]$$

$$= 1 + \mathbb{E}\left[\sum_{t=1}^{T} \hat{c}_t \cdot \left(p_t - p_\gamma^\star\right)\right]$$

$$\leq R(p_\gamma^\star) - R(p_1) + \eta \mathbb{E}\left[\sum_{t=1}^{T} \sum_{i=1}^{|\Pi|} p_t(i) \hat{c}_t(i)^2\right]$$

$$\leq \frac{1}{v}\Phi(p_\gamma^\star) - \frac{1}{\eta}H(p_1) + \eta \mathbb{E}\left[\sum_{t=1}^{T} \sum_{i=1}^{|\Pi|} p_t(i) \hat{c}_t(i)^2\right]$$

$$\leq 1 + \frac{|\Pi| \log(|\Pi|T)}{v} + \frac{\log |\Pi|}{\eta} + \eta \mathbb{E}\left[\sum_{t=1}^{T} \sum_{i=1}^{|\Pi|} p_t(i) \hat{c}_t(i)^2\right],$$

as claimed. □

*Proof of Theorem 6.* Using Theorem 7 and considering the stability term, for all $t \in [T]$ and $i \in [|\Pi|]$ it holds that

$$\mathbb{E}_t\left[\hat{c}_t(i)^2\right] = \mathbb{E}_t\left[\left(\sum_{y \in \pi_i(x_t)} \frac{\ell_t(y)\mathbf{1}\{y \in a_t\}}{Q_t(y)}\right)^2\right]$$

$$= m^2 \mathbb{E}_t\left[\left(\frac{1}{m}\sum_{y \in \pi_i(x_t)} \frac{\ell_t(y)\mathbf{1}\{y \in a_t\}}{Q_t(y)}\right)^2\right]$$

$$\leq m\mathbb{E}_t\left[\sum_{y \in \pi_i(x_t)} \left(\frac{\ell_t(y)\mathbf{1}\{y \in a_t\}}{Q_t(y)}\right)^2\right]$$

$$= m \sum_{y \in \pi_i(x_t)} \frac{\ell_t(y)^2}{Q_t(y)},$$

where the third line uses Jensen's inequality, and the last equality uses the fact that $\mathbb{E}_t[\mathbb{1}\{y \in a_t\}] = Q_t(y)$. Thus, the stability term is bounded by

$$\eta\mathbb{E}\left[\sum_{t=1}^{T}\sum_{i=1}^{|\Pi|} p_t(i)\mathbb{E}_t\left[\hat{c}_t(i)^2\right]\right] \leq \eta m\mathbb{E}\left[\sum_{t=1}^{T}\sum_{i=1}^{|\Pi|} p_t(i) \sum_{y \in \pi_i(x_t)} \frac{\ell_t(y)^2}{Q_t(y)}\right]$$

$$= \eta m\mathbb{E}\left[\sum_{t=1}^{T}\sum_{y=1}^{K}\underbrace{\sum_{i=1}^{|\Pi|} p_t(i)\mathbb{1}\{y \in \pi_i(x_t)\}}_{Q_t(y)} \frac{\ell_t(y)^2}{Q_t(y)}\right]$$

$$= \eta m\mathbb{E}\left[\sum_{t=1}^{T}\|\ell_t\|_2^2\right] \leq \eta smT,$$

and plugging in the specified values for $\eta$ and $\nu$ gives the desired bound. $\qquad\square$

## C  Open Problems

**Full-bandit feedback.**  Our results for PAC learning in the semi-bandit setting raise a natural question regarding the *full-bandit* feedback model, which is characterized by the fact that the observation upon predicting $a_t \in \mathcal{A}$ consists of a single number $r_t^\top a_t$. Specifically, it is unclear given our results whether adapting to reward sparsity and obtaining improved sample complexity bounds of the form $\text{poly}(s, m)/\varepsilon^2$ is possible in the full-bandit model. While we do not fully know how to extend our techniques to the full-bandit setting, we present some interesting ideas which may provide initial directions towards answering this question. We focus on the question of how we should modify the log-barrier potential given in Eq. (1) to accommodate full-bandit feedback, that is, such that its gradient would correspond to the variance of an appropriate full-bandit reward estimator. Indeed, importance-weighted reward estimators for combinatorial bandits are widely used in the literature [see, e.g., 3], and it is straightforward to construct such an unbiased estimator $R(\pi)$ and show that its variance can be bounded as

$$\text{Var}[R(\pi)] \leq s^2 \cdot \pi(x)^\top C^{-1}\pi(x),$$

where $C$ is the covariance matrix associated with sampling a policy from $p$. Interestingly, the latter quantity can be shown to be proportional to the gradient of the following log-determinant potential:

$$f(p; x) \triangleq -\log\det(C(p; x)) = -\log\det\left(\sum_{k=1}^{|\Pi|} p(k)\pi_k(x)\pi_k(x)^\top\right),$$

which would seem like a natural generalization of the log-barrier potential used in Eq. (1). This variance bound, however, contains no dependence on the reward function $r$, and in turn the existing analysis would result in a sample complexity of $\approx K/\varepsilon^2$, which we would like to avoid.

**Improving the dependence on $K$.**  A natural question which is also left open in [17] for the single-label setting concerns improving the dependence on $K$ in the additive term of the sample complexity bound. We strongly believe that this term could be significantly improved, potentially reaching $K/\varepsilon$ which would result in a minimax optimal sample complexity bound. The high polynomial degree in the current bound mostly stems from requiring the approximate minimization of $F(\cdot)$ as a means to obtain a point at which the gradient is small. Intuitively, minimizing the objective $F(\cdot)$ itself is not our primary goal, but rather we are interested in finding a point at which the gradient of $F(\cdot)$ is small. This naturally leads us to look for an optimization procedure which would minimize the norm of the gradient of $F(\cdot)$ directly, which can be done in various unconstrained optimization problems [see, e.g., 21], using a sample complexity which only depends logarithmically on the objective's smoothness. The constrained nature of our problem of interest, though, poses a significant challenge to adopting this approach directly. Nevertheless, we conjecture that leveraging specific properties of the objective function (such as self-concordance) may enable extending these methods to constrained settings. Such results would not only improve the sample complexity bounds for bandit multiclass classification but could also be of independent interest to the convex optimization research community.

**Extension to infinite classes.** Another natural research direction is to extend our result given in Theorem 1 to hypothesis classes which could be infinite, but exhibit certain properties which allow for replacing the dependence on $\log |\mathcal{H}|$ with some combinatorial dimension. For the single-label setting, [17] show that a finite Natarajan dimension is indeed such a property. We thus expect that for list classification, an appropriate generalization of the Natarajan dimension would give such results, in a similar manner to how the Littlestone dimension is generalized by [39] to list classification in the online setting.

