# OpenReview forum: "From Contextual Combinatorial Semi-Bandits to Bandit List Classification: Improved Sample Complexity with Sparse Rewards"
_NeurIPS.cc/2025/Conference — NeurIPS 2025 poster_

### Official Review · Reviewer_FW6h · 2025-07-01

**Clarity:** 4
**Significance:** 3
**Originality:** 3
**Rating:** 4
**Confidence:** 3

**Summary:**

This paper investigates the $(\epsilon,\delta)$-PAC learnability of multiclass list classification with semi-bandit feedback. In this model, at each round, the nature generates an instance $X$ and a list of $s$ labels $Y = \{y_1,y_2,\ldots,y_s\}$, where $y_i \in [K]$, from a distribution $D$ and shows the instance $X$ to the learner. Then, the learner predicts a list of $m$ labels $\hat{Y}$ and gets the feedback $\hat{Y} \cap Y$. After $T$ rounds, the learner will output a concept $h \in \mathcal{H}$ that maps the instance $X$ to a list of $m$ labels. They investigate the sample complexity of learning the $\epsilon$-optimal hypothesis with the probability of at least $1-\delta$ and get an upper bound of $\tilde{O}((\text{poly}(K/m)+sm/\epsilon^2)\log(|\mathcal{H}|/\delta))$, when the concept class $\mathcal{H}$ is finite. They also provide an $\Omega(sm/\epsilon^2)$ lower bound.

For the $s=m=1$ case, they get a $O((K^7+1/\epsilon^2)\log(|\mathcal{H}|/\delta))$ upper bound. For the regret minimization setting, they get an expected regret bound of $\tilde{O}(|\mathcal{H}|+\sqrt{smT\log|\mathcal{H}|})$ for the adversarial setting.

**Questions:**

In the contextual combinatorial semi-bandit problem, your algorithm works under the case that the reward function is additive, that is, $r(A\cup B) = r(A) + r(B)$. What if the reward function is submodular, or more generally, any set function? Is it possible to get the same sample complexity?

**Ethical Concerns:**

["NO or VERY MINOR ethics concerns only"]

**Final Justification:**

I think this problem is reasonable and meaningful from a theoretical perspective. The result is solid, but under some common assumptions. This paper also focuses on the case where the concept class is finite. Therefore, I give this paper a borderline accept.

**Limitations:**

Yes

**Quality:**

3

**Strengths And Weaknesses:**

Strength:
1. The PAC learnability of multiclass classification with bandit feedback is an interesting and important problem. There are significant connections between this problem and the long-standing multi-armed bandit problem. This paper considers multiclass list classification with semi-bandit feedback, and the result can be applied to solve the contextual combinatorial semi-bandit problem. Their algorithm also provides the regret bound for the contextual combinatorial semi-bandit problem.
2. The algorithm provided by this paper is computationally efficient with an efficient ERM oracle. Therefore, this result may be interesting in both the statistical learning theory community and the broader TCS community.
3. The presentation of this paper is quite good.

Weakness:

The result only works for finite hypothesis classes. It is not clear what will happen when the hypothesis class is infinite.
The assumption that every call of the ERM oracle only spends constant time is not that benign when there is an infinite hypothesis class.

---

> ### Author Rebuttal · Authors · 2025-07-30
>
> Thank you for your review. ​​We hope that our responses below will fully address your comments and questions.
>
> You actually seem to appreciate the contributions of the paper, and in particular we are glad that you find the research question interesting and important, and that our results could be of interest to the broader TCS community. We have to ask that, if you tend to favor seeing the paper accepted, please consider upgrading your score. If any concern still remains, we will be glad to try and clarify further during the discussion period.
>
>
> > “The result only works for finite hypothesis classes. It is not clear what will happen when the hypothesis class is infinite”
>
> While extending our results for infinite classes would definitely be interesting, the question of optimal sample complexity rates for infinite hypothesis classes is not well understood even in the simpler full-information version of the problem (see e.g. the discussion in [10,38]). That considered, characterizing the optimal rates in the more general bandit setting still seems quite formidable and falls beyond our scope in this paper.
>
> Further note that even in the simpler single-label setting studied in [17], the upper bound provided for classes with finite Natarajan dimension is not known to be rate-optimal. We thus chose to focus on finite classes as a natural first and crucial step in studying the more challenging bandit version of the problem.
>
> > “The assumption that every call of the ERM oracle only spends constant time is not that benign when there is an infinite hypothesis class”
>
> Our assumption that each call to the ERM oracle takes constant time was made solely for simplicity of presentation. If instead the computational cost of the ERM oracle to $\mathcal{H}$ is denoted by $T_{ERM}(\mathcal{H})$, the overall computational complexity of our PAC learner would be $O((K/m)^5) \cdot T_{ERM}(\mathcal{H})$.
>
> Such computational assumptions about the oracle’s runtime are standard in the literature (see e.g. [1,2]), as a means to avoid additional structural assumptions and to allow for arbitrary hypothesis classes. We will revise our presentation to make all of this clear - thanks.
>
> **Questions:**
>
> > “What if the reward function is submodular, or more generally, any set function?”:
>
> You raise a very interesting question. In combinatorial semi-bandits, the reward functions exhibit an inherent structure as sums of the reward values of the $K$ basic actions, and in particular exhibit the additivity property which you mentioned. Our current approach in the semi-bandit setting heavily relies on this additive structural property as is the case with other methods proposed in the semi-bandit literature, and we do not know whether or not our results can be generalized to wider classes of reward functions such as submodular functions. All of this could make up a wonderful direction for future research, in bandit list classification but also in combinatorial bandits more generally!
>
> **References:**
>
> [1] Dudik, Miroslav, et al. "Efficient optimal learning for contextual bandits." arXiv preprint arXiv:1106.2369 (2011).
>
> [2] Agarwal, Alekh, et al. "Taming the monster: A fast and simple algorithm for contextual bandits." International conference on machine learning. PMLR, 2014.
>
> [17] Erez, Liad, et al. "Fast rates for bandit PAC multiclass classification." Advances in Neural Information Processing Systems 37 (2024): 75152-75176.

---

> > ### Comment · Reviewer_FW6h · 2025-08-04
> >
> > Thanks a lot for your response. I understand that all those assumptions are common, and it is very hard to discuss the learnability problem when the concept class is infinite. But I still think this paper is borderline.

---

> > > ### Author Response · Authors · 2025-08-05
> > >
> > > Thank you for the response - we appreciate your perspective.  If there are any further concerns worth discussing, please keep us posted and we will gladly do so.

---

> ### Comment · Area_Chair_Rw1x · 2025-08-03
>
> Dear Reviewer FW6h,
>
> Please provide your feedback to the authors as early as possible after reading the rebuttal and other reviews, so there is enough time for back-and-forth discussion.  Thank you!
>
> Best,
> Area Chair

---

### Official Review · Reviewer_f4D1 · 2025-07-02

**Clarity:** 1
**Significance:** 2
**Originality:** 3
**Rating:** 2
**Confidence:** 3

**Summary:**

The paper studies the bandit multiclass list-classification problem, an extension of bandit multiclass classification when multiple labels can be predicted for each example. The paper considers how to build an algorithm that aims to approximate the best classification hypothesis out of a finite hypothesis class $|\mathcal{H}|$. Formally, at each time $t$, the environment draws a sample $(x_t,y_t) \sim \mathcal{D}$, after observing $x_t$, the algorithm predicts a set of labels and receives feedback indicating which of the predicted labels were correct. The paper draws a strong connection between this classification problem and the contextual combinatorial semi-bandit (CCSB) framework, which is leveraged to design the algorithm to approximate the best hypothesis with a small number of samples. A sample complexity of $\mathcal{O} \left ( \left (\frac{K^9}{m^8} + \frac{sm}{\epsilon^2} \right )\log \frac{|\Pi |}{\delta} \right )$ is then proven for this algorithm. The paper then provides a lower bound which matches their sample complexity asymptotically for small $\epsilon$. In the appendix, the paper provides an analysis of a similar setting with an objective of regret minimization and provides an upper bound on the regret of $\mathcal{O} \left ( \sqrt{sm T \log |\Pi|} \right ) $.

**Questions:**

1. What do you think is the limiting factor to applying the current types of results to infinite hypothesis classes?

2. In the CCSB setting, and in algorithm 1, the main problem at hand is to choose the right policy $\pi \in \mathcal{\Pi}$. In the framework of the paper, my understanding is that it is related to the hypothesis class $\mathcal{H}$, is that correct? Could you clarify that? Furthermore, it feels like in the paper, you sometimes switch notation from the ones of the classification problem to the CCSB. It is my understanding that the underlying assumption is that there is an equivalence between the two problem. I feel that having one set of homogeneous notation would make navigating the paper easier.

3. Algorithm 1 uses a procedure that is based on two separate phases. The first phase explores uniformly, while the second one uses the previously acquired information to focus the exploration. Is there a particular reason for having two phases (in contrast with having an algorithm that starts by exploring uniformly but performs a focused exploration similar to phase 2 gradually, while incrementally refining its estimates )?


4. In the paper, you do not provide any assumption regarding the space $\mathcal{X}$. It is my understanding that this comes at no additional cost from a discrete space $\mathcal{X}$ because you assume that the hypothesis classes are finite. Essentially, using the fact that both $\mathcal{H}$ and $\mathcal{Y}$ are finite, the problem reduces to one with finite $\mathcal{X}$. Does that seem correct?

**Ethical Concerns:**

["NO or VERY MINOR ethics concerns only"]

**Final Justification:**

My main concern is the following: While they state it as a theorem, no formal proof is provided for the lower bound. While they provide a proof sketch, I believe it is not sufficient. Without any formal proof, this is, in my opinion, mostly a claim.

The two other points are secondary. I detail them below.

I do believe their argument that CCSB is more general than BMLC is true. For now, there is no proof or sketch on how to reduce BMLC to CCSB but the authors mentioned to improve the transition between problems, which i believe should clear that up. My point for the issue in formulating the result in terms of CCSB is more related to the lack of clear translation between the notations used in both settings. As the results are stated with notation for the CCSB problem and no clear correspondence is provided, it is not straightforward to even read the results for BMLC. If the interesting problem to make the paper compelling is indeed BMLC, it is not great to rely on an implicit reduction to present the results.

I believe their result is an extension of [1], where after reducing list-classification to multiclass classification (on a bigger class set resulting for the combinatorial set of lists of classes) the similarity between these combinatorial classes is leveraged. While this surely does not straightforwardly follow, even only for notational complexity and because a proper setting is required, the result is not very surprising and does not answer a very compelling and general question.

[1] Erez, L., Peled-Cohen, A., Koren, T., Mansour, Y., & Moran, S. (2024). Fast rates for bandit PAC multiclass classification. Advances in Neural Information Processing Systems, 37, 75152-75176.

**Limitations:**

yes

**Paper Formatting Concerns:**

No major formatting concerns.

in algorithm 1 you mention "delta distribution", it wouldn't hurt to add one line to write what it means

line 227- 228, i believe it shouldn't be $[0,1]^K$ but ${0,1]}^K$

**Quality:**

1

**Strengths And Weaknesses:**

Overall, the paper proposes a technically interesting algorithm for list-based classification in a bandit setting, but a clearer exposition and complete proofs are required. Furthermore, this work would benefit from broader generalization.

 # Strengths

- The paper is well-written and describes the related literature clearly.

- The paper provides lots of insight into the motivation and key elements of their algorithm, as well as the main technical elements necessary to prove the results.

 # Weaknesses

- The paper provides lower bounds for the problem but does not include a proof. While some very high-level intuition is provided, the proof does not appear in the appendices. Considering this, it is difficult to consider Theorem 2 as more than an intuition. This makes it difficult to gauge whether their approach is optimal or if more progress can be expected.

- While the introduction and setting showcase the fact that the paper provides both a sample complexity and regret approach, only the sample complexity approach is addressed in the main paper. Consider presenting both the sample complexity and regret minimization settings (the algorithms and main results) in the main paper to clarify the full scope of contributions. If space is an issue, consider reducing the amount of technical details in the main paper and stick to high-level ideas and/or main differences with existing techniques, and provide further insight in the appendix.

- The paper does not address the case of infinite hypothesis classes. Providing this kind of result, even with restricting assumptions on the classes of hypotheses, would greatly improve the value of the contribution.

- Some literature was missed on combinatorial bandits with sparse rewards. Please consider extending the related work, especially since you claim to provide new results in that line of work. ( see [1] for instance)

- The paper mostly uses the notation introduced when describing the Contextual Combinatorial Semi-Bandit setting. I understand that the algorithm used is based on CCSB literature, and I appreciate the fact that the link is explicitly stated, as it makes understanding the whole thing easier, yet it is counterintuitive to introduce the problem you are trying to solve and to switch notations right away afterwards. Furthermore, using different notation in the algorithms makes it difficult to follow. See, for instance, question 2.


[1] Perrault, P., Valko, M., & Perchet, V. (2020, July). Covariance-adapting algorithm for semi-bandits with application to sparse outcomes. In Conference on Learning Theory (pp. 3152-3184). PMLR.

---

> ### Author Rebuttal · Authors · 2025-07-30
>
> Thank you for your review. We hope that our responses below will alleviate your concerns and allow you to reconsider your low score. We are happy to answer any further questions during the discussion period.
>
> > ”The paper provides lower bounds for the problem but does not include a proof…”:
>
> While we do not provide a formal proof of the lower bound, converting the high-level intuition to a formal proof can be done using techniques which are standard in the bandits literature, see e.g. Theorem 5.1 in [1]. That being said, we do agree that including a formal proof would be of value, and we will do so in the final version.
>
> > “While the introduction and setting showcase the fact that the paper provides both a sample complexity and regret approach, only the sample complexity approach is addressed in the main paper”:
>
> Indeed, our primary focus is on the PAC version of the problem, which also contains the bulk of the technical contributions of our work. We defer the regret minimization section to the appendix in order to allow space for providing more context and details on our contributions in the PAC setting. Space permitting, we will definitely include the result for the online setting in the main text of the final version.
>
> > ”The paper does not address the case of infinite hypothesis classes. Providing this kind of result, even with restricting assumptions on the classes of hypotheses, would greatly improve the value of the contribution”;
>
> While extending our results for infinite classes would definitely be interesting, the question of optimal sample complexity rates for infinite hypothesis classes is not well understood even in the simpler full-information version of the problem (see e.g. the discussion in [10,38]). That considered, characterizing the optimal rates in the more general bandit setting still seems quite formidable and falls beyond our scope in this paper.
>
> Further note that even in the simpler single-label setting studied in [17], the upper bound provided for classes with finite Natarajan dimension is not known to be rate-optimal. We thus chose to focus on finite classes as a natural first and crucial step in studying the more challenging bandit version of the problem.
>
> > “Some literature was missed on combinatorial bandits with sparse rewards”:
>
> Thank you for pointing out [1] to us, we will make sure to include and discuss it in our related work section. We remark however that [1] focuses on the special case of linear contextual bandits, which is quite different from our more general setting where the underlying mappings from contexts to rewards may be arbitrary (but bounded).
>
> If you are aware of any additional references that we may have missed, we would appreciate it if you could let us know so that we can include them in the revision.
>
> **Questions:**
>
> > “1. What do you think is the limiting factor to applying the current types of results to infinite hypothesis classes?”:
>
> See our response above regarding infinite hypothesis classes.
>
> > “2. …it feels like in the paper, you sometimes switch notation from the ones of the classification problem to the CCSB. It is my understanding that the underlying assumption is that there is an equivalence between the two problem.”:
>
> There is **no** equivalence between the two problems: the bandit list classification problem is, directly by definition, a special case of the more general contextual combinatorial semi-bandit (CCSB) setting. There is no assumption that needs to be made for this to hold. Since we are mainly dealing with algorithms, in most of the paper we focus on the more general CCSB problem and use the associated terminology. You are correct to note that hypotheses $h \in \mathcal{H}$ in the classification setting are then analogous to the policies $\pi \in \Pi$ in CCSB.
>
> In any case, we will do our best to improve the clarity of our notation in the revision.
>
> > “3. Algorithm 1 uses a procedure that is based on two separate phases… Is there a particular reason for having two phases”:
>
> You raise an interesting question. We are currently not sure whether a “fully adaptive” exploration method could result in better sample complexity bounds, and we ultimately chose to separate the algorithm into two phases as it makes the analysis conceptually simpler by breaking down the main result into two stages: first optimizing for a low-variance exploration distribution and then using this distribution to estimate rewards more effectively.
>
> > “4. …you do not provide any assumption regarding the space $\mathcal{X}$... Essentially, using the fact that both $\mathcal{H}$ and $\mathcal{Y}$ are finite, the problem reduces to one with finite $\mathcal{X}$. Does that seem correct?”
>
> This is **not** correct in general: $\mathcal{X}$ could be infinite even when $\mathcal{H}$ is finite. Perhaps you meant that if $\mathcal{X}$ and $\mathcal{Y}$ are both finite, then the effective number of hypotheses (cardinality of $\mathcal{H}$) is finite? (This is a correct statement, since the number of possible mappings from $\mathcal{X}$ to $\mathcal{Y}$ is finite in this case.)
>
> In any case, the cardinality of the space $\mathcal{X}$ **plays no role** at any point in the analysis, only the size of $\mathcal{H}$ does. This should not come as a surprise: even in classical classification settings such as binary classification, the cases where $\mathcal{H}$ is finite while $\mathcal{X}$ is possibly infinite are interesting well-studied regimes, and the complexity of the problem is determined by the cardinality/dimension of $\mathcal{H}$, regardless of the size of $\mathcal{X}$.
>
> **Typos:**
>
> Thank you for bringing these to our attention, we will make sure to correct them.
>
>
> **References:**
>
> [1] Perrault, P., Valko, M., & Perchet, V. (2020, July). Covariance-adapting algorithm for semi-bandits with application to sparse outcomes. In Conference on Learning Theory (pp. 3152-3184). PMLR.
>
> [10] M. Charikar and C. Pabbaraju. A characterization of list learnability. In Proceedings of the 55th Annual ACM Symposium on Theory of Computing, pages 1713–1726, 2023.
>
> [17] Erez, Liad, et al. "Fast rates for bandit PAC multiclass classification." Advances in Neural Information Processing Systems 37 (2024): 75152-75176.
>
> [38] S. Moran, O. Sharon, I. Tsubari, and S. Yosebashvili. List online classification. In The Thirty Sixth Annual Conference on Learning Theory, pages 1885–1913. PMLR, 2023.

---

> > ### Comment · Reviewer_f4D1 · 2025-08-03
> >
> > Thank you for the response. It did made the contribution and setting overall clearer.
> >
> > I am still a bit confused about the fact that the paper is presented as solving the bandit multiclass list-classification problems. Yet all the results are stated for another problem. While bandit multiclass list-classification can be cast as an instance of CCSB, I believe it would be clearer to choose one setting and stick to it.

---

> > ### Comment · Reviewer_f4D1 · 2025-08-05
> > **Question for infinite hypothesis classes**
> >
> > I am wondering about what limits your ability to obtain infinite hypothesis classes. While I understand that obtaining optimal rates would be very challenging, do you think it would be possible to still be possible to get some results?
> >
> > As I said, I believe results for the infinite hypothesis class would be very interesting. Getting any results regardless of their optimality, would therefore be valuable.

---

> > > ### Author Response · Authors · 2025-08-05
> > >
> > > Thank you for your message. We are glad that our contribution and setting are now clearer.
> > >
> > > - Regarding list-classification vs CCSB: Our main goal in the paper is indeed solving the bandit multiclass list-classification problem, and we do that by solving a *strictly more general problem* - the contextual combinatorial semi-bandits (CCSB) problem. We do understand, however, that the transition between different terminologies used in these distinct problems may be confusing in a single paper, and we will make sure to improve this in the final version.
> > >
> > > - Regarding the challenge in addressing infinite hypothesis classes: this is something we are currently thinking about; the core difficulty is that we do not know what is the appropriate “dimension” concept in the context of bandit list classification. Answering this question is fundamental to designing learning algorithms - as much as it is the case, for example, in the classical PAC learning setting, where, once the correct (VC) dimension is determined, the corresponding learning algorithm immediately follows. As we alluded to above, this is not even understood in the simpler, full-information version of the list classification problem.
> > >
> > > - Regarding whether it is possible to get non-optimal, finite sample-complexity results for infinite hypothesis classes: this is actually a great question that we need to think more about; currently we only have in this general case finite sample-complexity results that scale exponentially with the list size, but we will definitely give this some more thought until the final version of the paper.

---

> > > > ### Comment · Reviewer_f4D1 · 2025-08-06
> > > >
> > > > Thank you for the answer.

---

> > > > > ### Author Response · Authors · 2025-08-07
> > > > >
> > > > > Thanks again for your engagement in the discussion thus far.  We would appreciate it if you could let us know - did our message above address your remaining concerns? Are any further clarifications needed?

---

> ### Comment · Area_Chair_Rw1x · 2025-08-03
>
> Dear Reviewer f4D1,
>
> Please provide your feedback to the authors as early as possible after reading the rebuttal and other reviews, so there is enough time for back-and-forth discussion.  Thank you!
>
> Best,
> Area Chair

---

### Official Review · Reviewer_KD2J · 2025-07-03

**Clarity:** 2
**Significance:** 3
**Originality:** 2
**Rating:** 4
**Confidence:** 3

**Summary:**

The paper tackles the setting of bandit multi-label list classification from a learning-theoretic perspective and analyze it under the agnostic PAC and agnostic online learning setting. In addition, the authors show that their results for the agnostic online learning setting can also apply in the combinatorial semi-bandit setting.

Multi-label list classification is understood as predicting a list of labels for each instance, and each instance is allowed to have multiple true labels. In the bandit version of this scenario (or semi-bandit as the authors note), the learner receives a response that explains which subset of the labels are correct but not necessarily the set of all true labels for that instance. The reward function is designed to be the count of number of true labels for a given instance that occur in the list predicted by the learner. In all settings considered, the hypothesis class $\mathcal{H}$ is assumed to be finite with a finite label space $\mathcal{Y}$ that has $|\mathcal{Y}|=K$.

In agnostic PAC learning, the authors develop an algorithm (Algorithm 1) that returns an $\epsilon$-optimal hypothesis with probability $1 - \delta$ with a sample complexity on the order of $O(\mathrm{poly}(\frac{K}{m}) + \frac{sm}{\epsilon^2})$ and show a lower bound with $\Omega(\frac{sm}{\epsilon^2})$. The main algorithmic technique consists of employing a learner to run a Frank-Wolfe procedure on a specially-designed convex potential whose gradient effectively estimates the variance of the reward estimator of any single policy. When the setting is simplified to single-label classification, the bound has an improved dependence on $K$ ($K^9$ to $K^7$) compared to previous work. In agnostic online learning, the authors devise an algorithm that has expected regret at most growing $O(\sqrt{KT})$ where $T$ is the number of rounds.

**Questions:**

- For the case of agnostic online learning, there doesn't seem to be a lower bound on the regret showing that the regret achieved by Algorithm $4$ is in fact optimal or loose. Can the authors elaborate on the optimality of the regret bounds in the case of agnostic online learning?

**Ethical Concerns:**

["NO or VERY MINOR ethics concerns only"]

**Final Justification:**

I thank the authors for answering all my questions with detailed explanations during the rebuttal period. I recommend that the authors make the necessary revisions to the paper to make the more readable. As a result, I retain my original rating of a 4 for this paper.

**Limitations:**

Yes

**Quality:**

3

**Strengths And Weaknesses:**

Strengths:
- This setting is analyzed in both agnostic PAC learning and agnostic online learning where upper and lower bounds are proven in the agnostic PAC setting.
- The paper generalizes their work to the setting of contextual combinatorial semi-bandits with a bound that directly relies on the sparsity of rewards rather than the size of the label set.
- The authors introduce a clever twist of constructing a convex potential whose gradient is exactly related to the variance of reward estimators. By employing the Franke-Wolfe procedure, the resulting mixed policy distribution has guarantees on the maximum value of the reward estimators allowing the algorithm to effectively estimate the reward estimators for all policies.

Weaknesses:
- While the usage of Franke-Wolfe leads to non-trivial sample complexity bounds, the use of Franke-Wolfe itself is not novel as prior work (citation [17]) employs a similar procedure in the case of single-label bandit classification. As a result, the innovation is more about tweaking the methodology rather than introducing a new optimization paradigm.
- While the authors do provide upper and lower bounds in the agnostic PAC setting showing the importance of the $\frac{sm}{\epsilon^2}$ term, the hidden polynomial factor $(K^9)$ is pretty large signifying that there is still a somewhat sizeable gap between the two bounds.
- The writing of the paper is at times hard to understand, especially for an audience not well-versed in this specific literature. The paper could benefit from having an improved exposition of the learning setting itself and provide a better proof sketch of Theorem $1$ that elaborates the core technical ideas in a clearer fashion.

---

> ### Author Rebuttal · Authors · 2025-07-30
>
> Thank you for your review---we hope that our responses below will fully address your concerns. Reading your comments, you seem to appreciate the contributions of the paper, and in particular we are glad that you found our technique of constructing the log-barrier potential clever. If you tend to favor seeing the paper accepted, it is crucial that you kindly consider upgrading your score. If any concern still remains, we will be glad to try and clarify further during the discussion period.
>
> > “While the usage of Franke-Wolfe leads to non-trivial sample complexity bounds, the use of Franke-Wolfe itself is not novel as prior work (citation [17]) employs a similar procedure in the case of single-label bandit classification.…”:
>
> While a variant of Frank-Wolfe was indeed used in [17] for the single-label setting, that paper employed a stochastic version (SPIDER-FW) that relies on variance-reduced gradient estimators. Instead, we use the more standard non-stochastic version of Frank-Wolfe, but on the empirical version of the potential, and then analyze its stochastic guarantees via novel Rademacher complexity arguments (see Appendix A.1). This modification not only introduces a different methodology, but also allows obtaining improved additive polynomial dependence on $K$ in the single-label setting.
>
> > “...there is still a somewhat sizeable gap between the two bounds.”:
>
> We agree that the additive $K^9$ upper bound can be significant when the number of labels is large. As you recognized, the main point is that this is an additive term and the main $1/\epsilon^2$ term is independent of $K$ (disregarding log factors). That said, it is indeed very interesting for us to improve this additive term, even just in the standard multi-class setting, but we suspect this might require additional techniques.
>
> > “The writing of the paper is at times hard to understand, especially for an audience not well-versed in this specific literature”:
>
> We appreciate the feedback. In the following revision, we will do our best in order to improve the clarity of the exposition, the problem setup and the proof sketch of our main result. If you have in mind any specific places where we should focus, we will be very glad to hear. Thanks!
>
> > “Can the authors elaborate on the optimality of the regret bounds in the case of agnostic online learning?”:
>
> While we do not prove a formal lower bound, the prior work [18] established matching lower bounds in the single-label setting (see lines 958-959 in the supplementary). Their construction can be extended in a straightforward manner to the list classification setting, which would provide a matching lower-order term for the list classification setting. We add that the main term of the form $\sqrt{smT \log |\Pi|}$ can be formally shown to be tight via arguments analogous to the lower bound given in Theorem 2 which we sketch in the main text. A formal lower bound for the online regret minimization setting was not included in the paper as this setting is not our main focus.
>
> **References:**
>
> [17] Erez, Liad, et al. "Fast rates for bandit PAC multiclass classification." Advances in Neural Information Processing Systems 37 (2024): 75152-75176.
>
> [18] Erez, Liad, et al. "The real price of bandit information in multiclass classification." The Thirty Seventh Annual Conference on Learning Theory. PMLR, 2024.

---

> > ### Comment · Reviewer_KD2J · 2025-08-04
> >
> > I thank the authors for providing a detailed response to my review.
> >
> > Point 1: The additional clarification of the differences in the usage of Frank-Wolfe are clearer to me. I believe the improved additive polynomial in $K$ is more of an application of the different methodology of Frank-Wolfe established in this paper rather than purely technical contribution on its own. As a result, I still stick to my original point.
> >
> > Point 2: I agree that this is an interesting open question to tackle.
> >
> > Point 3: Thank you for being open to revisions. I have a couple of suggestions that may improve the paper:
> > 1. I think it would be nice to have a dedicated paragraph to describing the difference between Franke-Wolfe used in [17] compared to how it's being used in this paper. This will clearly establish the different challenges and contributions.
> > 2. It might be worth it to begin Section 2 with the contextual combinatorial semi-bandits since the results are expressed in that setting. And then it can be expanded on how it can be seen from the perspective bandit multiclass list classification.
> > 3. In Section 3.1, I struggled a bit in understanding the relationship between the LOO and ERM oracles. It would be nice to have a high-level intuitive picture of a connection between these two so it's more understandable.
> >
> > Question 1: Thank you for addressing the question on regret lower bounds.
> >
> > I thank the authors for addressing all my questions and concerns. Given the author response and my response and original review, I will retain my original score of 4 for this paper.

---

> > > ### Author Response · Authors · 2025-08-05
> > >
> > > Thank you for your message and the additional feedback. We are indeed open to suggestions, and we will consider all of those while preparing the final version of the paper.
> > >
> > > Regarding the relationship between the LOO and ERM oracles: the linear optimization oracle (LOO) is standard in the context of the Frank-Wolfe method, while the ERM oracle is commonly used in classification and contextual bandit scenarios, which is the main reason for the different terminology we used for the two different types of oracles. As it turns out, the LOO required by our application of Frank-Wolfe is a special case of the ERM oracle which we assume we have access to in the bandit multiclass list classification problem. We will make sure to outline a clearer connection between the two in the final version - thanks again for this and your other suggestions.

---

> ### Comment · Area_Chair_Rw1x · 2025-08-03
>
> Dear Reviewer KD2J,
>
> Please provide your feedback to the authors as early as possible after reading the rebuttal and other reviews, so there is enough time for back-and-forth discussion.  Thank you!
>
> Best,
> Area Chair

---

### Official Review · Reviewer_ZLpc · 2025-07-03

**Clarity:** 2
**Significance:** 2
**Originality:** 2
**Rating:** 3
**Confidence:** 3

**Summary:**

This paper studies multiclass list classification where the learner only receives semi-bandit feedback. That is, it assumes for each object $x$, there exist a set of $s$ true labels. Upon receiving $x$, the learner is to predict a list of $m$ labels and receives all correct labels among the $m$ labels. The paper establishes a PAC learning variant of the problem and proposes an algorithm with sample complexity of $\tilde{O}(\text{poly}(K/m)+sm/\epsilon^2)\log(\lvert{H}\rvert/\delta)$, improving on its previous work on a special case when $m=s=1$. The paper also considers a regret minimization setting with adversaries, and establishes a regret bound of $\tilde{O}(\lvert{H}\rvert+\sqrt{smT\log{\lvert{H}\rvert}})$.

**Questions:**

Can the authors give a more clear vision on why a PAC learning framework for the proposed problem is necessary?
If the multiclass list classification with bandit (or semi-bandit) feedback is not explored before, why considering adversarial data in this paradigm is necessary in this study?
How important the ERM oracle is? What if such oracle does not exist? If without such an oracle, is bandit multiclass list learning possible?

**Ethical Concerns:**

["NO or VERY MINOR ethics concerns only"]

**Final Justification:**

This paper studies a contextual combinatorial semi-bandit problem, and designs an algorithm with improved sample bound when the problem is restricted to a special cases of multiclass learning problem. When viewing the problem in a learning/classification scenario, it is very close to multilabel classification. The reviewer feels the reduction of the bandit problem to a traditional learning/classification problem was less natural. However, after discussing with the authors and area chair, the reviewer can appreciate the proposed algorithm that works both for bandit and multilabel problems with improved sample complexity, hence changing my recommendation from rejection to a borderline.

**Limitations:**

The authors shall discuss the necessity of the ERM oracle.

**Paper Formatting Concerns:**

No concerns.

**Quality:**

3

**Strengths And Weaknesses:**

First of all, I have a question on whether it is necessary to pose this problem in the multiclass and list learning paradigms, especially with (semi)-bandit feedback, plus adversarial data and online learning scenarios. It feels too much effort to fit the problem in the PAC learning setting. Given the motivative application mentioned in this paper, it feels much more like a recommendation system problem or contextual combinatorial semi-bandit problems. Moreover, the algorithm assumes having access to an ERM oracle, which is also less a PAC learning paradigm. I am not convinced that this is a reasonable and well-posed problem. The paper is generally hard to read, with many terminology remain unexplained.

---

> ### Author Rebuttal · Authors · 2025-07-30
>
> Thank you for your review. We hope that our responses below will alleviate your concerns and will allow you to reconsider your score. We would be glad of course to answer any further questions during the discussion period.
>
> > “Can the authors give a more clear vision on why a PAC learning framework for the proposed problem is necessary? ”; “It feels too much effort to fit the problem in the PAC learning setting… it feels much more like a recommendation system problem or contextual combinatorial semi-bandit problems”
>
> **We do in fact formalize the bandit list classification problem as a contextual semi-bandit problem**, but with an additional crucial structure: reward sparsity, that abstracts the fact there is only a small number of correct labels per example. By referring to this as a “PAC setting” we mean that this is a stochastic i.i.d. setup and the guarantees we establish are $\epsilon$-approximate with high probability $\geq 1-\delta$, independently of the data distribution. This is a customary jargon in the learning theory literature.
>
> Further, previous work on both single-label and list multi-class classification [14,15,17] also focused on what we call here the “PAC framework”, so it is only natural to consider the list-bandit version of the problem in this framework as well.
>
> >  “If the multiclass list classification with bandit (or semi-bandit) feedback is not explored before, why considering adversarial data in this paradigm is necessary in this study?”
>
> Note that in the PAC framework, which is our main focus, the data (examples and labels / contexts and rewards) is assumed to be **stochastic i.i.d.**, and **not adversarial** as you seemed to suggest. Since existing research on combinatorial semi-bandits also studies the setting of regret minimization with adversarial data, we also include additional results for the latter setting which we defer to the appendix/supplementary.
>
> > “the algorithm assumes having access to an ERM oracle, which is also less a PAC learning paradigm…How important the ERM oracle is? What if such oracle does not exist?”
>
> Assuming access to an ERM oracle is **only necessary for obtaining computationally efficient algorithms** (that is, with runtime polynomial in $\log |\mathcal{H}|$ and not in $|\mathcal{H}|$).
> The rest of our claims, namely the sample complexity bounds, hold regardless---but the algorithm that achieves them is in general not efficient. Indeed, in the absence of an ERM oracle, in the finite hypothesis class setting we focus on, the algorithm can optimize over the class directly by enumerating over all hypothesis, which would result in a time complexity that scales with $|\mathcal{H}|$ which we do not consider as “efficient”.
>
> We further note that in the vastly studied stochastic contextual bandit setting, such an ERM oracle assumption is standard in the literature (see e.g. [1,2]). Even in the classic (full-information) PAC learning setting, learning reduces in general to performing ERM over the hypothesis class which, without further structural assumptions, can only be done efficiently assuming access to an optimization/ERM oracle.
>
> > “The paper is generally hard to read, with many terminology remain unexplained.”
>
> We believe our paper’s structure, jargon and level of detail is in accordance with existing literature in this area of research. That said, we would appreciate it if you could point out any specific unexplained terminology so that we can learn where we can perhaps include further explanation. Thank you.
>
>
> **References:**
>
> [1] Dudik, Miroslav, et al. "Efficient optimal learning for contextual bandits." arXiv preprint arXiv:1106.2369 (2011).
>
> [2] Agarwal, Alekh, et al. "Taming the monster: A fast and simple algorithm for contextual bandits." International conference on machine learning. PMLR, 2014.
>
> [14] Daniely, Amit, and Shai Shalev-Shwartz. "Optimal learners for multiclass problems." Conference on Learning Theory. PMLR, 2014.
>
> [15] Daniely, Amit, et al. "Multiclass learnability and the ERM principle." J. Mach. Learn. Res. 16.1 (2015): 2377-2404.
>
> [17] Erez, Liad, et al. "Fast rates for bandit PAC multiclass classification." Advances in Neural Information Processing Systems 37 (2024): 75152-75176.

---

> ### Comment · Area_Chair_Rw1x · 2025-08-03
>
> Dear Reviewer ZLpc,
>
> Please provide your feedback to the authors as early as possible after reading the rebuttal and other reviews, so there is enough time for back-and-forth discussion.  Thank you!
>
> Best,
> Area Chair

---

> ### Comment · Reviewer_ZLpc · 2025-08-05
> **Official Comment by Reviewer ZLpc**
>
> Thank you for the response. Based on discussion, I think this problem is less natural in a learning setting, as traditional learning problems study functions with only one distinct output. As Reviewer f4D1 also noticed, it is more natural to formulate this problem in one setting (in this case, maybe better in CCSB) and stick to it.

---

> > ### Author Response · Authors · 2025-08-05
> >
> > Thank you for your message. Two quick comments:
> >
> > - Traditional learning problems indeed study hypotheses with only one output, but *this is exactly the novelty in the recent (and very active) line of work on list classification*, which our paper belongs to. The list setting is in fact natural in certain scenarios - we discuss the example of recommendation systems in our introduction.
> >
> > - Our main goal in the paper is solving the bandit multiclass list-classification problem, and *we do that by solving a strictly more general problem* - the contextual combinatorial semi-bandits (CCSB) problem. We do understand, however, that the transition between different terminologies used in these distinct problems may be confusing in a single paper, and we will make sure to improve this in the final version, as per your suggestions.

---

### Decision · Program_Chairs · 2025-09-17

**Decision:**

Accept (poster)

**Comment:**

The paper tackles bandit list multiclass classification where in each round, the learner receives a context and choose a set of m labels. The ground truth has s labels, and the learner observes the correctness of the label they choose.  The paper casts this as a contextual combinatorial semi-bandit (CCSB) problem and design two algorithms for this problem: one with stochastic reward (PAC setting) and one with adversarial reward (regret setting).

During the author-reviewer discussion phase, reviewers ZLpc, KD2J, and f4D1 raised clarity concerns. First, the switch between bandit list multiclass classification and CCSB is confusing for Reviewer ZLpc and f4D1: the paper could be framed just under the CCSB, and make bandit list multiclass classification as an important application, but the paper first states the latter and transitions to the former.  As f4D1 noted, the notation switch is too quick. Second, as KD2J pointed out, the paper is hard to read at times --- lacking sufficient high-level explanation for the algorithm and the proofs.

Based on my reading, technically, this looks like a standard contribution in a conference like NeurIPS, though not ground-breaking given the previous results of Erez et al's "Fast rates for bandit PAC multiclass classification" and "The real price of bandit information in multiclass classification".  The paper did make algorithmic and analysis advance in obtaining explorative distribution and low-variance reward estimator: for the potential function used to obtain low-variance estimator, they directly optimize the empirical version of the potential (instead of using an stochastic optimization procedure as in prior work), and use Rademacher complexity argument to establish uniform convergence. This allows an improved sample complexity bound, even in the single-label special case considered by [Erez et al, 2024]. As far as I can see, similar observations could be useful in other bandit problems involving solving a regularized objective.  Another contribution is the importance weight technique specific for the multilabel setting.

After author-reviewer and AC-reviewer discussions, there is still no consensus. The concerns of ZLpc and f4D1 still remain. After deeper discussions with them on these concerns, I conclude that the concerns are relatively minor compared to the main contribution. ZLpc connects the "PAC setting" in the paper to the "list PAC learning" framework in [7,10], which seems to be an overinterpretation in my opinion. ZLpc perhaps also overlooks the fact that list classification and multi-label classification are interchangeable ideas. The main concern of f4D1 is the missing formal proof of the lower bound.  However, it seems to me that the lower bound is a minor part of the contribution, and the proof sketch seems to be standard enough (except for the argument "in order to find an eps-optimal subset, Alg must identify at least half of the arms in S, and thus is required to estimate all of the arms’ rewards eps/2m" which definitely require additional explanation) to be converted to a formal proof.  Another concern of f4D1 is the novelty.  However, the reviewer did not provide more concrete evidence to support this concern.

Overall, I appreciate the main contribution --- the first sparsity-adaptive bound for contextual bandits, and other analysis/algorithmic contributions made by this work, so still recommend acceptance despite the negative evaluations by ZLpc and f4D1. However, the revised version of this paper needs to provide much more explanations than the current version for readers to easily understand it. Also, the authors should definitely write a formal proof for the lower bound.